# Understanding LoRA as Knowledge Memory: An Empirical Analysis

**Seungju Back** [* 1]   **Dongwoo Lee** [* 1]   **Naun Kang** [2]   **Taehee Lee** [2]   **S. K. Hong** [2]   **Youngjune Gwon** [2]   **Sungjin Ahn** [1 3]

## Abstract

Continuous knowledge updating for pre-trained large language models (LLMs) is increasingly necessary yet remains challenging. Although inference-time methods like In-Context Learning (ICL) and Retrieval-Augmented Generation (RAG) are popular, they face constraints in context budgets, costs, and retrieval fragmentation. Departing from these context-dependent paradigms, this work investigates a parametric approach using Low-Rank Adaptation (LoRA) as a modular knowledge memory. Although few recent works examine this concept, the fundamental mechanics governing its capacity and composability remain largely unexplored. We bridge this gap through the first systematic empirical study mapping the design space of LoRA-based memory, ranging from characterizing storage capacity and optimizing internalization to scaling multi-module systems and evaluating long-context reasoning. Rather than proposing a single architecture, we provide practical guidance on the operational boundaries of LoRA memory. Overall, our findings position LoRA as the complementary axis of memory alongside RAG and ICL, offering distinct advantages. Code and datasets are available at link.

## 1. Introduction

Large Language Models (LLMs) have enabled strong natural language understanding and generation across various domains such as mathematics, code, and scientific reasoning (Romera-Paredes et al., 2024; Chen et al., 2021; Tu et al., 2024; Wei et al., 2022). Despite this progress, an LLM's knowledge is largely fixed at pretraining time, making continuous knowledge updates a critical yet unsolved challenge—whether for incorporating new facts, domain-specific documents, or personalized information. A natural approach is to inject new knowledge via fine-tuning, but this is often impractical at scale: full retraining or supervised fine-tuning risks catastrophic forgetting (Kirkpatrick et al., 2017) and incurs large costs for updates (Dong et al., 2024).

To address these issues, a common alternative is to provide new information at inference time without modifying model parameters. In-Context Learning (ICL) (Brown et al., 2020) injects knowledge directly into the prompt, but it is constrained by the context window and the quadratic computation cost of processing long sequences. Retrieval-Augmented Generation (RAG) (Lewis et al., 2020) relies on similarity-based retrieval and chunking, which can fragment evidence under fixed context budgets and has limitations (Weller et al., 2025).

Meanwhile, Low-Rank Adaptation (LoRA) (Hu et al., 2022) is a widely used fine-tuning technique that freezes the pre-trained model and injects trainable low-rank matrices. While LoRA is typically used for task or domain adaptation, its efficiency and modularity suggest a second use: treating LoRA modules as *knowledge memory* that can be trained, swapped, and composed to update an LLM without full retraining. However, it is not obvious whether an adapter for task or domain adaptation also serves as a reliable store for precise, declarative knowledge.

Although recent works have begun utilizing LoRA as knowledge storage (Su et al., 2025; Zweiger et al., 2025; Caccia et al., 2025), these works do not prove LoRA's ability as knowledge memory, because they focus on performance gain in specific problem settings. They happen to include LoRA as a component of a pipeline, rather than systematically examining LoRA's intrinsic memory properties. Hence, the observed gains remain difficult to account for, raising doubts about the fundamental capabilities and limitations of LoRA as a general solution for knowledge memory—this motivates systematic investigation into its failure modes and deployment conditions. In fact, our results reveal that LoRA can catastrophically fail in certain settings, which necessitates careful configuration and practical guidance for practitioners—such as answers to the following questions. Can LoRA reliably memorize and retrieve factual knowledge? Does it scale to large knowledge bases, and if so, how should capacity be managed? Can LoRA-

---

[1]KAIST [2]Samsung SDS [3]New York University. Correspondence to: Sungjin Ahn <sungjin.ahn@kaist.ac.kr>.

*Proceedings of the 43rd International Conference on Machine Learning*, Seoul, South Korea. PMLR 306, 2026. Copyright 2026 by the author(s).

based memory complement or even replace non-parametric methods like RAG and ICL? If it serves only partial roles, under what conditions should it be deployed?

The goal of this work is to address these gaps by shifting the focus from system building to a systematic audit of LoRA as a parametric memory. We conduct a fine-grained empirical study to map the design space of LoRA, treating it not as a proven tool but as an object of investigation in its own right. To this end, we organize our study as a series of research questions around four dimensions: characterizing storage capacity, optimizing knowledge internalization within a single module, scaling to multi-LoRA systems, and conducting case studies that probe long-context reasoning and the role of hybrid parametric/non-parametric memory in realistic settings. To facilitate these investigations, we introduce PhoneBook and PaperQA, two novel benchmarks designed to probe specific memory properties.

Our findings suggest that LoRA memory is rarely a standalone solution, but it can serve as a practical and complementary option for knowledge updates alongside RAG and ICL. In particular, it tends to work best when used selectively or paired with non-parametric memory, while its effectiveness depends critically on supervision design and careful modular composition. Taken together, our study isolates the intrinsic properties and failure modes of LoRA as a knowledge memory and offers empirical guidance on when and how it should be used in practice.

## 2. Related Work[1]

**LLM Memory.** Extending LLM memory beyond temporary mechanisms like In-Context Learning (ICL) (Brown et al., 2020) is a central challenge. RAG (Lewis et al., 2020) externalizes knowledge via retrieval, but it is constrained by embedding-based similarity limits (Weller et al., 2025) and by top-$k$ selection under a fixed context budget, which can fragment evidence and hinder holistic use of long documents (Kuratov et al., 2024). Beyond retrieval, prior approaches span writable/readable non-parametric stores (Xu et al., 2025; Chhikara et al., 2025) and parametric or architectural mechanisms that internalize updates in weights or structure (Wang et al., 2024; Behrouz et al., 2024; 2025; Dao & Gu, 2024). These lines collectively leave open how to characterize the reliability and limits of compact parametric memory units under long-context, budgeted inference.

**LoRA.** LoRA (Hu et al., 2022) was introduced for parameter-efficient adaptation and is now commonly trained, stored, and swapped across tasks or domains (Ostapenko et al., 2024a; Li et al., 2025; Pletenev et al., 2025; Liang et al., 2025). Recent work explores LoRA as a module for knowledge memory, including distillation-based objec-

---

[1]Detailed related works can be found in Appendix A.

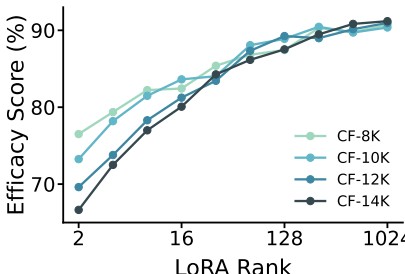

*Figure 1.* Performance trend on the CF as rank increases.

tives that align an adapter to an in-context teacher (Caccia et al., 2025) and meta-learning pipelines that optimize synthetic supervision for self-updating (Zweiger et al., 2025). However, these methods primarily emphasize end-to-end system for a single module, leaving the memory-specific questions of what LoRA can store, when it saturates, and how supervision format affects factual retrievability less pinned down. Related analyses of LoRA efficiency and optimization (Schulman & Lab, 2025) largely target general adaptation dynamics, rather than memory-centric limits and long-context behavior.

**Multi-LoRA Composition and Routing.** To scale beyond a single adapter, prior work combines multiple LoRAs via interpolation/merging or routing-based selection (Huang et al., 2023; Feng et al., 2024; Dou et al., 2024; Prabhakar et al., 2025; Fleshman & Van Durme, 2025c). Notably, Parametric-RAG style systems retrieve and merge document-specific LoRAs (Su et al., 2025; Tan et al., 2025), but their end-to-end behavior depends on coupled choices (how modules are produced, retrieved, and merged), and practical questions such as how many modules can be merged and under what interference regimes are not fully characterized. Meanwhile, several composition methods target task/skill adapters and rely on learned or optimized mixing weights (Huang et al., 2023; Prabhakar et al., 2025), which may be mismatched to plug-and-play knowledge settings where combinations change per query and must remain training-free at inference. Overall, existing work motivates a clearer understanding of when modular composition helps for factual storage and when routing/merging becomes a bottleneck.

## 3. Experimental Setup

Our empirical analysis spans a broad range of configurations across research questions. Due to space constraints, we present a comprehensive master configuration table in Appendix D. We refer readers to this appendix for the specific settings corresponding to each experiment.

## 4. Characterizing LoRA's Memory Ability

Our initial investigation aims to uncover fundamental properties of a single LoRA module as a memory unit. How

does memorization scale with LoRA rank and the amount of new knowledge, and where does it saturate? To this end, we study two controlled benchmarks: PhoneBook (PB; Appendix B) and CounterFact (CF) (Meng et al., 2022).

Following prior work that uses controlled synthetic benchmarks to study LLM knowledge and generalization (Lampinen et al., 2025; Allen-Zhu & Li, 2024; Jiang et al., 2024), we use two programmatically scalable datasets with minimal overlap to pretraining. PB is a key–value dataset of fictional name–phone-number pairs, and CF consists of counterfactual edits that conflict with pre-trained beliefs (e.g., "Paris is in Italy"). By varying the number of inserted items/edits (reporting tokenized size), PB probes storage of arbitrary associations (e.g., user IDs or database entries) while CF probes targeted revision of established facts (e.g., updating outdated entity information).

We evaluate PB using exact match and evaluate the efficacy score for CF. We sweep LoRA ranks from 2 to 1024 and vary the knowledge size from 1K to 20K tokens on Llama-3.1-8B (Grattafiori et al., 2024) and Qwen3-8B (Yang et al., 2025a) (with Llama tokenizer).

## Q1. Is Memory Capacity Scalable with LoRA Rank?
To test whether LoRA's memory capacity scales with rank, we train modules with varying ranks under a fixed data setting and evaluate memorization on both PB and CF. We consistently observe that increasing rank improves memorization: for example, on CF ranging from 8K to 14K length, efficacy rises steadily with rank (Figure 1). This indicates that rank effectively controls the parameter budget available for storing new knowledge (details and full results in Appendix F).

> **Implication:** LoRA capacity increases with rank, making rank a practical knob for trading off memory capacity against parameter cost.

## Q2. Is There a Finite Capacity Limit? How Is It Determined?
We probe LoRA's capacity by increasing the knowledge load while holding rank fixed. We observe a rank-dependent saturation pattern: at a fixed rank, performance drops as the number of stored items grows, and lower ranks degrade earlier. Figure 2 shows that higher-rank modules maintain higher performance over larger loads, indicating a higher effective capacity under the same training setup (details in Appendix G).

> **Implication:** Under a fixed training setup, LoRA exhibits a finite capacity, with rank acting as a key knob that shifts the saturation point.

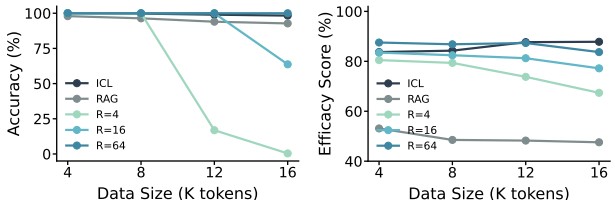

*Figure 2.* Performance as data size increases. **(Left)** PB results. **(Right)** CF results.

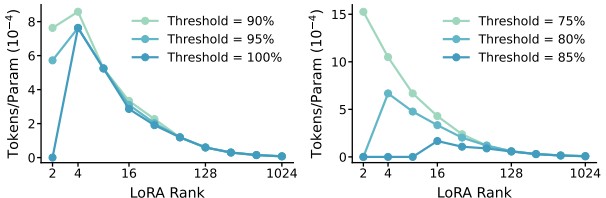

*Figure 3.* Efficiency as rank increases. **(Left)** PB results. **(Right)** CF results.

## Q3. Is Using the Highest Rank Always the Most Efficient Choice?
While Q1–Q2 show that higher ranks yield greater absolute capacity, a practical question remains: is maximizing rank the most efficient choice? Higher ranks incur substantial costs in parameters, training time, and memory, motivating a measure of parameter efficiency—the amount of knowledge stored per trainable parameter. Concretely, we estimate the largest tokenized knowledge load $T_{\max}$ for which performance stays above a fixed threshold $\tau$ (i.e., before saturation; Q2), and normalize it by the number of trainable parameters:

$$\text{Efficiency} = \frac{T_{\max}}{N_{\text{params}}}. \quad (1)$$

We find that the highest rank is not the most efficient. Figure 3 shows non-monotonic efficiency curves that peak at specific ranks and then decline; across both PB and CF, peak efficiency occurs at low ranks, indicating diminishing returns beyond a certain rank (details in Appendix H).

> **Implication:** A trade-off exists between absolute memory capacity and parameter efficiency in LoRA modules. Simply maximizing rank can lead to resource waste.

**Discussion.** We draw three takeaways about LoRA as a memory unit. First, capacity scales with rank, supporting LoRA as a controllable parametric memory. Second, capacity is finite, making memory budgeting unavoidable. Third, when measured by parameter efficiency, lower ranks can be more efficient, showing a capacity-cost trade-off. These findings motivate two directions:

1. **Optimizing a single LoRA.** Since LoRA has limited capacity, the key is to store information in a high-density, non-redundant form. This calls for data and supervision

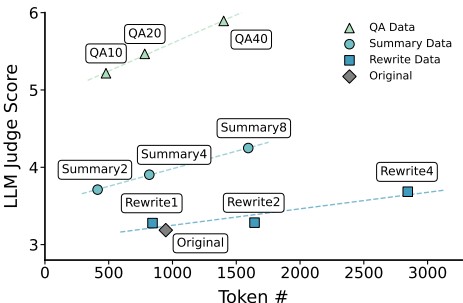

Figure 4. Performance scaling with different synthetic data generation methods.

Table 1. Performance with average token counts across different synthetic data combinations.

| Methods | Llama-3.1-8B | Qwen3-8B | Tokens |
|---|---|---|---|
| Original | 3.187 | 4.587 | 947 |
| QA40 | 5.893 | 5.409 | 1,401 |
| Original + QA40 | 6.300 | 5.371 | 1,841 |
| Summary8 + QA40 | 6.380 | **5.662** | 2,995 |
| Rewrite4 + QA40 | 6.650 | 5.427 | 3,004 |
| Original + Summary8 + Rewrite4 + QA40 | **6.822** | 5.614 | 4,088 |

designs that maximize memory utilization, which we study in Section 5.

2. **Scaling via Multi-LoRA systems.** The efficiency of low-rank modules suggests partitioning knowledge across many small adapters. However, scaling to many modules introduces system-level bottlenecks—notably misrouting and parameter interference during merging—which we analyze in Section 6.

# 5. Optimizing a Single LoRA

**The PaperQA Benchmark.** To evaluate a LoRA module's ability to internalize and reason over *new, complex* information, we introduce a novel benchmark, PaperQA. PaperQA is constructed from 15 recent papers (NeurIPS 2024, ICLR 2025, ICML 2025) to reduce the risk of pretraining contamination. It comprises a total of 450 QA pairs organized in a hierarchical suite spanning (i) key information recall, (ii) contextual comprehension, and (iii) logical structure inference. We score responses with a rubric-based LLM judge, enabling graded evaluation beyond exact-match accuracy. See Appendix C for dataset statistics, construction details, and prompts.

**Q4. How Does Synthetic Data Enhance Single LoRA Memorization?** Since a LoRA module has finite capacity, effective memory formation depends on how densely the training data encodes the target knowledge. Although prior work suggests synthetic supervision can improve knowledge learning in LLMs (Lampinen et al., 2025; Park et al., 2025;

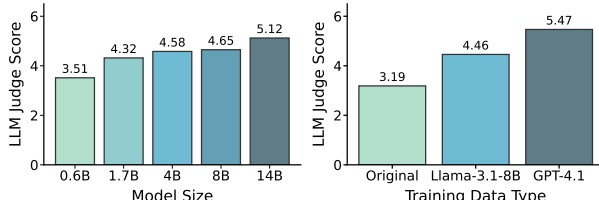

Figure 5. (**Left**) Performance across different Qwen3 model sizes. (**Right**) Performance comparison when using Llama vs. GPT for synthetic data generation.

Lin et al., 2025; Yang et al., 2025b; Zhang et al., 2024b), its behavior when applied to LoRA under increasing data budgets and the relative merits of different formats remain underexplored. We therefore compare three synthetic formats—**QA**, **Summary**, and **Rewrite**—against a raw-text baseline, scaling the amount of training data (number of generated instances) made using GPT-4.1 (Achiam et al., 2023) generation.

Figure 4 shows two consistent trends. First, all synthetic formats substantially outperform raw text. Second, performance improves monotonically as we scale the amount of synthetic supervision. Among them, QA achieves the strongest gains and the best token-efficiency, suggesting that task-aligned, high-density supervision can yield more improvement per training token (details in Appendix I).

> **Implication:** Converting raw documents into structured synthetic supervision improves LoRA memorization and also scales effectively with additional training data.

**Q5. What Are the Effects of Combining Synthetic Data Formats?** Having found QA to be a strong single format, we next test whether adding other supervision signals further improves memorization. We construct mixed training sets by concatenating QA with other data formats, thereby increasing the amount and diversity of supervision (Table 1). Across both models, combining formats generally improves over QA alone: on Llama, the most comprehensive mixture achieves the best score, while on Qwen the best result is obtained by Summary+QA, with the full mixture remaining competitive. These results suggest that supplementing the model with a diverse view of the same content can provide useful extra training signals (details in Appendix J).

> **Implication:** Combining diverse synthetic data formats yields complementary benefits, enabling LoRA modules to surpass the performance ceilings of single format data.

**Q6. How Does the Scale of Base Models Affect LoRA Knowledge Internalization?** We examine how base model scale influences LoRA's knowledge internalization, a key factor for balancing performance and cost. To this end, we fine-tuned LoRA on Qwen3 (Yang et al., 2025a) mod-

els ranging from 0.6B to 14B with identical training data and performed a per-model hyperparameter search to select strong configurations for each scale. Figure 5 (left) shows performance improved with model size but non-linearly: significant gains from 0.6B-1.7B and 8B-14B, minimal improvement from 1.7B-8B. Unlike ICL, which leverages base model reasoning, LoRA primarily stores knowledge in its added parameters, making it less sensitive to base model capacity in certain ranges (details in Appendix K).

> **Implication:** Larger base models improve LoRA performance, but non-linearly with diminishing returns in mid-range scales (1.7B-8B).

**Q7. Does Synthetic Data Generator Quality Impact LoRA Performance?** Following Q4 and Q5's findings on synthetic data benefits, we investigate whether generator model quality affects outcomes—a practical trade-off between costly API models and local alternatives. We compare LoRA trained on QA data from GPT-4.1 versus Llama-3.1-8B using the same pipeline. Figure 5 (right) shows that GPT-4.1-generated data consistently yielded higher performance than Llama-3.1-8B data. Superior generators produce more accurate, logically sound, and comprehensive training data, with these quality differences directly transferring to final LoRA performance (details in Appendix L).

> **Implication:** Stronger data generators yield better knowledge internalization in LoRA.

## 6. Scaling to Multi-LoRA Systems

Section 4 showed that a single LoRA module has finite, rank-dependent capacity and that low-rank adapters can be highly parameter-efficient. Motivated by these observations, we next explore scaling memory by partitioning knowledge across multiple LoRA modules. However, the benefit of modularity depends on system-level choices—how knowledge is partitioned, how relevant LoRA is routed at test time, and how multiple LoRA modules are composed under routing uncertainty. We aim to provide a systematic characterization of these design dimensions by (i) establishing an oracle upper bound with perfect routing, (ii) quantifying degradation under practical routing, and (iii) evaluating merging strategies that trade off robustness and interference.

**Q8. Can Multiple Small LoRAs Outperform a Single Large LoRA?** Motivated by the high parameter-efficiency of low-rank adapters (Section 4), we ask whether partitioning a fixed parameter budget across many small LoRAs can increase total memory capacity. To isolate the effect of partitioning from system effects, we use PhoneBook with 64K tokens and compare three settings under a matched

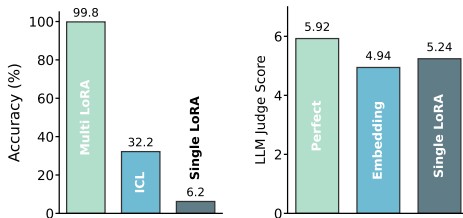

*Figure 6.* **(Left)** Multi-LoRA PoC on 64K PhoneBook. **(Right)** Performance gap between perfect and RAG routing.

trainable-parameter budget: (1) full-context ICL, (2) a single large LoRA trained on the entire dataset, and (3) multiple small LoRAs, each trained on a disjoint partition. For multi-LoRA, we assume an oracle router that always activates the correct module, yielding an upper bound where routing error is removed.

Figure 6 (left) shows that ICL degrades under long context and that a single LoRA saturates at this knowledge load. In contrast, multi-LoRA maintains high accuracy by distributing the data across modules and activating the appropriate partition at test time. This result establishes that, given accurate module selection, partitioning can convert a fixed parameter budget into substantially higher effective memory capacity, motivating routing as the next bottleneck (details in Appendix M).

> **Implication:** Under oracle routing, distributing a fixed parameter budget across multiple small LoRAs can outperform a single large LoRA by avoiding single module saturation.

**Q9. How Much Performance Loss Does a More Practical Routing Incur?** While Q8 established an oracle upper bound, multi-LoRA systems in practice require routing under imperfect signals. Because PhoneBook is a structured key–value task that does not naturally align with standard text-embedding retrieval, we use PaperQA, where each LoRA module stores the contents of a single paper, and routing reduces to matching a question to its relevant document.

We compare three settings: (1) perfect routing (oracle selection), (2) embedding-based routing, a standard method in RAG, that selects the top-1 module by text embedding similarity, and (3) a single LoRA baseline that stores all papers in one module. Figure 6 (right) shows that practical routing incurs a substantial drop relative to the oracle and can even underperform single LoRA, indicating that misrouting can negate the benefit of partitioning by activating an irrelevant module. This gap is consistent with known limitations of embedding-based retrieval, which can fail to represent the information needed to identify the truly relevant source for a given query (Weller et al., 2025). In Appendix N, we further compare embedding-based rout-

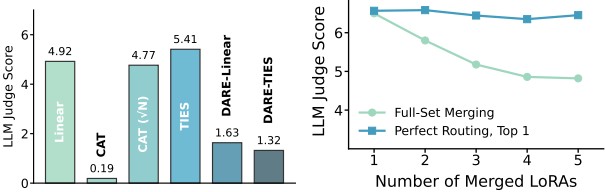

*Figure 7.* **(Left)** Merging strategies comparison. **(Right)** Comparison between selecting a single optimal LoRA and merging N LoRAs.

ing with token-based routers, including Arrow (Ostapenko et al., 2024b), SpectR (Fleshman & Van Durme, 2025b), and LAG (Fleshman & Van Durme, 2025a); the results suggest that routing remains a bottleneck across router families, with no token-based method consistently eliminating the gap.

> **Implication:** Routing accuracy is a primary bottleneck for multi-LoRA; mitigating routing uncertainty is essential for realizing the gains suggested by oracle routing.

**Q10. Can Merging Multiple LoRAs Mitigate Routing Error?** Given routing uncertainty, a natural hedge is to retrieve multiple candidate modules (top-3 in our experiments) and merge them, rather than committing to a single (possibly misrouted) LoRA. We evaluate five representative merging strategies used in prior work: (1) linear averaging as a simple baseline, (2) concatenation (CAT) as used in modular systems such as PRAG (Su et al., 2025), (3) scaled concatenation (CAT-$1/\sqrt{N}$), which rescales each LoRA factor by $1/\sqrt{N}$, (4) TIES (Yadav et al., 2023) for interference-aware adapter merging via parameter pruning, and (5) DARE (Yu et al., 2024), which merges by dropping and rescaling delta parameters.

Figure 7 (left) shows that TIES is the most robust choice: it substantially improves over the routed multi-LoRA baseline and reaches performance comparable to the single LoRA model, indicating that appropriate merging can partially compensate for misrouting. Linear averaging is a strong baseline but is generally less robust than TIES, while DARE underperforms, suggesting that stochastic dropping may discard critical information in this memorization setting. Finally, vanilla CAT collapses because summing injects an $N$ times over-perturbation; rescaling by $1/\sqrt{N}$ corrects this and recovers most of the gap to Linear, identifying scale mismatch as the dominant failure mode. (details in Appendix O).

> **Implication:** Interference-aware merging (e.g., TIES) can mitigate routing errors, making multi-LoRA more robust than selecting a single retrieved module.

**Q11. How Does the Number of Merged LoRAs Affect Performance?** Choosing how many modules to merge (top-$N$) mediates a trade-off in multi-LoRA systems: larger $N$ can hedge against misrouting, but may amplify interference when adapters are composed. To isolate the effect of merging itself, we remove routing errors by merging the ground-truth set of $N$ modules corresponding to the documents from which each query is drawn, and then vary $N$ from 1 to 5. Following Q10, we use TIES for merging.

As shown in Figure 7 (right), performance peaks at $N=1$ and then decreases monotonically as $N$ grows. This trend indicates that even when all necessary knowledge is included, merging additional modules can progressively dilute or interfere with the stored information, quickly degrading answer quality (details in Appendix P).

> **Implication:** Increasing the number of merged LoRAs can rapidly degrade performance due to interference, so multi-module systems must balance recall against merge robustness rather than blindly increasing $N$.

# 7. A Case Study on Long, Complex Data

Our previous experiments used controlled benchmarks (e.g., PhoneBook, CounterFact, and PaperQA) to isolate core factors in LoRA-based memory, including single module capacity, supervision efficiency, and system bottlenecks in routing and merging. We now extend these findings to a different operating regime: long-form documents where evidence is distributed across distant spans and answering a query requires cross-chunk synthesis. In this setting, the challenge is not only how much knowledge a LoRA can store, but also how well modularized memories preserve coherence when each module observes only a partial view, and multi-hop dependencies can span multiple chunks.

To probe this regime, we conduct a case study on NarrativeQA (NQA) (Kočiský et al., 2018) and QuALITY (Pang et al., 2022). Rather than optimizing for peak leaderboard performance, we use these benchmarks as a stress test to understand (i) when partitioned LoRA memory breaks under long-context, multi-hop reasoning, (ii) how combining LoRA with external context (ICL/RAG) restores performance, and (iii) whether LoRA-based memory can offer practical efficiency advantages under repeated querying.

We evaluate four base models: Llama-3.2-1B, Llama-3.1-8B, Qwen3-1.7B, and Qwen3-8B. For multi-LoRA, we train one LoRA per chunk and use a retrieval-based router to select top-1 or top-3 modules, which are then merged with TIES. We report ROUGE-L on NQA and accuracy on QuALITY, and compare against full-context ICL, standard RAG, and KM$_{SDCD}$ (Caccia et al., 2025). Full dataset details are provided in Appendix Q. To further stress-test

*Table 2.* Performance comparison on NarrativeQA and QuALITY across different models.

| Method | Llama-3.2-1B | | Llama-3.1-8B | | Qwen3-1.7B | | Qwen3-8B | |
|---|---|---|---|---|---|---|---|---|
| | NarrativeQA | QuALITY | NarrativeQA | QuALITY | NarrativeQA | QuALITY | NarrativeQA | QuALITY |
| *Closed Book* | | | | | | | | |
| Base model | 9.08 | 25.75 | 13.08 | 32.01 | 13.27 | 26.42 | 16.35 | 26.06 |
| KM$_{SDCD}$ | 23.32 | **30.21** | **29.07** | 30.21 | 19.45 | 28.73 | **28.68** | 33.67 |
| Single LoRA | **23.81** | 29.86 | 27.05 | **42.42** | **24.78** | 33.15 | 25.78 | 44.10 |
| Multi-LoRA Top1 | 16.87 | 24.32 | 19.95 | 37.83 | 15.90 | 32.71 | 19.13 | 39.46 |
| Multi-LoRA Top3 | 16.85 | 26.43 | 22.42 | **42.42** | 16.27 | **39.93** | 21.15 | **46.26** |
| *Open Book* | | | | | | | | |
| ICL | 24.52 | 26.01 | 33.81 | **70.20** | 27.34 | 46.80 | 35.22 | 72.89 |
| RAG | 21.90 | 26.12 | 29.20 | 63.79 | 23.80 | 43.43 | 31.15 | 64.53 |
| Single LoRA + ICL | 24.73 | 28.34 | 35.39 | 64.12 | 28.37 | 46.59 | 32.34 | 67.49 |
| Single LoRA + RAG | 24.12 | 29.86 | 32.18 | 42.42 | 27.12 | 33.15 | 31.37 | 44.10 |
| Multi-LoRA Top1 + ICL | 20.57 | 27.23 | 33.62 | 58.69 | 19.09 | 43.48 | 24.48 | 64.34 |
| Multi-LoRA Top1 + RAG | 19.22 | 28.90 | 25.03 | 47.26 | 19.10 | 38.40 | 24.42 | 48.14 |
| Multi-LoRA Top3 + ICL | **26.41** | 30.00 | **38.78** | 69.22 | **29.17** | **56.52** | 37.23 | **76.26** |
| Multi-LoRA Top3 + RAG | 24.53 | **31.20** | 35.55 | 60.34 | 29.10 | 54.99 | **37.64** | 64.31 |

whether LoRA-based memory survives in more extreme long-context regimes, we additionally evaluate on 100K+ token benchmarks, ∞Bench (Zhang et al., 2024a) and Long-Bench v2 (Bai et al., 2025), with results summarized in Appendix Q.

**Q12. How Does LoRA Memory Perform on Long-Context Multi-Hop QA Task?** In the closed-book setting on NarrativeQA, multi-LoRA underperforms a single LoRA across all four base models (Table 2), indicating that for multi-hop QA, the gains from partitioning are often offset by system-level noise. We attribute this mainly to the compounding effects of routing and merging (Q9–Q10), and to chunk-induced boundaries that make cross-segment synthesis difficult.

On QuALITY, the gap is smaller: Top-3 multi-LoRA is often competitive and can outperform single LoRA for some models, suggesting that tasks with more localized evidence are less sensitive to these boundaries (details in Appendix R).

> **Implication:** While modularity offers scalability, it introduces structural boundaries; when queries require cross-chunk synthesis, routing and merging noise can outweigh the benefits of partitioned memory.

**Q13. Can LoRA Perform Better When Used with External Context?** Given the limitations of multi-LoRA's fragmented memory, we test whether supplying external context at inference can help. We build hybrid systems by pairing single LoRA and multi-LoRA with ICL or RAG, using either the top-1 or top-3 retrieved modules. These are benchmarked against standalone ICL, RAG, and closed-book LoRA baselines. As shown in Table 2, adding external

context consistently enhances performance. The gains are most substantial for multi-LoRA, where context compensates for the precision loss inherent in merging modules. Notably, ICL yields a greater performance uplift than RAG. This suggests that the continuous, global context supplied by ICL is uniquely effective at restoring the narrative cohesion lost among the fragmented LoRA modules, a weakness that snippet-based retrieval from RAG cannot fully overcome (details in Appendix S).

> **Implication:** LoRA achieves stronger performance when paired with external context, outperforming standalone LoRA, RAG, or ICL. This shows that LoRA can be effective as a complementary parametric memory rather than a substitute.

**Q14. Can Merging LoRAs Improve Contextual Continuity?** Beyond relying on external context to maintain continuity in multi-LoRA systems, we investigate the direct synthesis of knowledge from multiple LoRA modules for multi-hop QA. Our empirical results in Table 2 demonstrate that merging the top-3 modules outperforms selecting the top-1, with the sole exception of the 1B model in the closed-book setting. This advantage is particularly pronounced when augmented with external context from ICL or RAG, indicating that the composition of specialized LoRA memories provides a more robust and comprehensive knowledge foundation than a single module (details in Appendix T).

> **Implication:** Merging multiple LoRAs acts as a mechanism for reconstructing the model's intrinsic contextual continuity, and this positive effect is amplified when combined with external context.

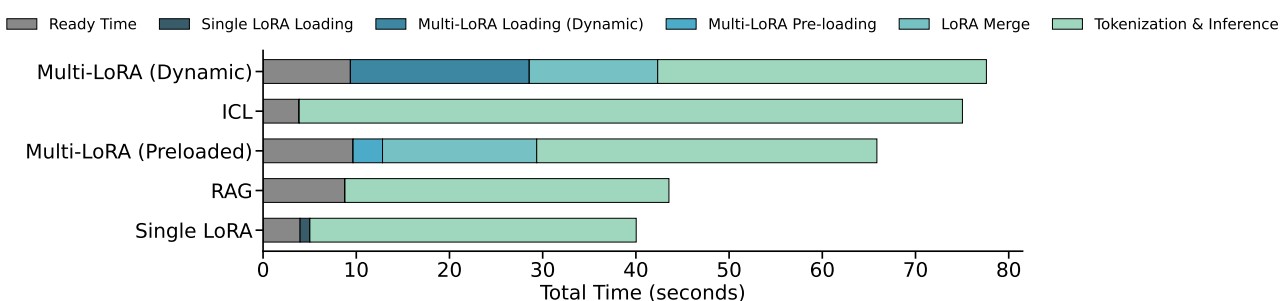

*Figure 8.* A breakdown of total processing time for different methods. Ready time denotes one-time setup before processing any query, including base model loading and, for retrieval-based methods, RAG initialization such as embedding-model loading, chunk embedding, and FAISS index construction.

**Q15. Can LoRA Save Time?** While previous sections established the performance of LoRA-based memory, practical viability also depends on computational efficiency. Context-based methods like ICL and RAG are computationally expensive, as they must process long context windows for every query. This analysis quantifies whether LoRA, by internalizing knowledge into its parameters, offers a more efficient inference alternative. We benchmarked the time to process 30 sequential questions from a single NarrativeQA document on a Llama-3.1-8B model.

Figure 8 illustrates the results. As hypothesized, LoRA-based methods exhibit shorter pure inference times by eliminating the need to process thousands of context tokens per query. However, they introduce overhead from loading and merging LoRA modules. For single LoRA, this is a minimal, one-time cost to load and attach the module. To mitigate the prohibitive I/O latency of dynamically loading modules in the multi-LoRA setup, we employ a pre-loading strategy where all document-relevant modules are loaded into GPU memory beforehand. Despite the remaining overhead of merging LoRAs for each query, our multi-LoRA pre-loading system achieves a lower total processing time than the ICL-based approach. The remaining overhead is largely systems-driven, leaving clear headroom for further optimization (details in Appendix U).

> **Implication:** In scenarios requiring repeated and inter-active access to a consistent knowledge base, the LoRA-based memory approach offers a substantial computational advantage over context-dependent methods.

**Additional results.** In the appendix, we further analyze (i) where to place Knowledge LoRA via layer/module ablations and (ii) whether LoRA variants improve memory. We find that applying LoRA to early layers and FFN generally yields stronger memorization and later saturation than attention-only or late-layer placement. For DoRA (Liu et al., 2024) and PiSSA (Meng et al., 2024), gains are mixed and model-/setting-dependent, with no consistent advantage over standard LoRA. See Appendix E for full results.

## 8. Conclusion

This work provides a systematic empirical characterization of Low-Rank Adaptation (LoRA) as a modular parametric knowledge memory for large language models. Across controlled synthetic settings and document-level benchmarks, we map the operational boundaries of LoRA memory—from single module capacity to multi-module bottlenecks—and distill actionable design principles for practice.

**Treat a Single LoRA as a Resource-Constrained Memory Unit.** A single LoRA exhibits *finite but scalable* storage: increasing rank raises the capacity ceiling, yet the gain is not proportional to parameter cost. Concretely, we observe non-monotonic parameter efficiency, where smaller ranks can store more usable knowledge per parameter than larger ranks. This suggests that memory-centric LoRA design should prioritize right-sized modules rather than defaulting to maximal rank.

**Maximize Knowledge Density via Synthetic Data.** Because LoRA memory is capacity-limited, how we store matters as much as how many parameters we allocate. We find that raw text is an inefficient substrate for memorization, whereas task-aligned, high-density synthetic formats are critical for effective internalization. Moreover, mixing complementary formats (QA, summaries, rewrites) yields consistent gains beyond any single format, highlighting the value of diversity in the memory-formation pipeline.

**Architect for Modularity, but Control System-Level Failure Modes.** Partitioning knowledge across multiple small LoRAs can unlock substantially larger total capacity under ideal routing; however, real systems are dominated by routing and merging errors. We quantify that imperfect routing can negate the benefits of modularization, and that merging introduces a sharp trade-off: increasing the number of merged modules raises recall but amplifies interference and dilutes the signal. Practically, scalable multi-LoRA systems require (i) stronger routing and (ii) interference-aware merging applied to a small set of high-confidence modules.

**Embrace Hybrid Memory for Robustness.** Our results in-

dicate that LoRA-based parametric memory is best viewed not as a replacement for context-based methods, but as a complementary component. While LoRA can reduce inference costs for repeated access to a stable knowledge base, peak performance on long and multi-hop settings is achieved when LoRA is combined with external context (ICL/RAG), which mitigates fragmentation and restores global coherence. Together, these findings can position LoRA as a practical complementary axis of memory alongside RAG and ICL, and motivate hybrid systems as a promising direction for robust and efficient knowledge access.

**Limitation and Future Work** Our study isolates the memory properties of LoRA modules in controlled settings, but does not constitute a full continual-update deployment. A key next step is to evaluate and design LoRA memory under time-incremental, topic-diverse streams (e.g., daily news updates), where new information is logically connected over time and must be stored, retrieved, and revised continuously. On the system side, scaling requires more robust routing and interference-aware merging for long-document multi-hop queries, as well as practical adapter serving (caching, prefetching, and kernel-level optimization) to manage the latency of multi-module composition.

# Acknowledgements

This research was supported by Samsung SDS. Seungju Back, Dongwoo Lee, and Sungjin Ahn were also supported by the Brain Pool Plus Program (No. 2021H1D3A2A03103645) and the GRDC (Global Research Development Center) Cooperative Hub Program (RS-2024-00436165) through the National Research Foundation of Korea (NRF), funded by the Ministry of Science and ICT (MSIT).

# Impact Statement

This paper presents work whose goal is to advance the field of machine learning by systematically characterizing LoRA as a parametric knowledge memory. While improving the practicality of modular knowledge injection could potentially be misused to distribute harmful content through adapters, we believe the ethical implications are consistent with those well established in the broader fine-tuning literature. We encourage practitioners to apply standard safeguards such as data governance and adapter auditing when deploying LoRA-based memory systems.

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

# A. Related Works in Detail

**LLM Memory.** A central challenge in continual learning is enhancing the memory capacity of large language models (LLMs). In-context learning (ICL) (Brown et al., 2020) offers a temporary mechanism but is constrained by window length. To overcome this, system-level solutions introduce non-parametric external memory modules that can be dynamically written and retrieved, as seen in Xu et al. (2025) and Chhikara et al. (2025). Conversely, parametric approaches aim to internalize knowledge directly into model weights; for instance, WISE (Wang et al., 2024) duplicates full network layers to handle sequential editing. Advancing further into long-context capabilities, strategies such as Behrouz et al. (2024; 2025); Dao & Gu (2024) attempt to extend memory through architectural adaptation. Distinguishing our work, we leverage LoRA to instantiate a parameter-efficient plug-and-play memory module for long-context scenarios, specifically focusing on quantifying the static capacity of this modular memory.

**Retrieval-Augmented Generation.** Retrieval-Augmented Generation (RAG) (Lewis et al., 2020) addresses memory limitations by retrieving relevant external documents and conditioning generation on them. RAG has proven highly effective for many tasks, but its embedding-based retrieval mechanism has inherent representational limitations (Weller et al., 2025). Moreover, top-$k$ retrieval restricts the accessible context, which poses challenges for tasks requiring holistic integration of information spread across an entire document (Kuratov et al., 2024).

**LoRA.** Low-Rank Adaptation (LoRA) (Hu et al., 2022) has become one of the most widely adopted parameter-efficient fine-tuning (Mangrulkar et al., 2022) (PEFT) methods. Beyond standard fine-tuning, recent works have explored leveraging LoRA within *continual learning* frameworks, where modular adapters are employed to facilitate incremental updates while mitigating catastrophic forgetting (Ostapenko et al., 2024a; Li et al., 2025) (Pletenev et al., 2025; Liang et al., 2025). While these approaches share the goal of task adaptation, more direct attempts to utilize LoRA as a knowledge memory module have recently emerged, such as SEAL (Zweiger et al., 2025) and DCD (Caccia et al., 2025).

Zweiger et al. (2025) proposed SEAL, a meta-learning framework where an LLM learns to generate its own synthetic training data via reinforcement learning. However, SEAL relies on a computationally expensive outer loop to train a generator LLM, often targeting much longer contexts, which limits its practical applicability. Similarly, Caccia et al. (2025) introduced DCD, which utilizes a Deep Context Distillation (DCD) objective to align a single LoRA with an in-context teacher model. This method also incurs significant computational overhead due to the distillation framework aligning hidden states across all layers. While our work shares the exploration of synthetic data generation strategies for LoRA training with these studies, we distinguish our approach by eliminating the computational overheads associated with SEAL's outer loop and DCD's distillation. Rather than focusing on proposing a specific training method for a *single* LoRA module, we provide a comprehensive analysis of LoRA's viability as a practical LLM memory, investigating critical properties including its fundamental capacity, data efficiency, and extendibility to multi-LoRA scenarios.

Concurrently, Schulman & Lab (2025) shares our interest in analyzing the optimized settings, capacity, and efficiency of LoRA; however, while they focus on the training dynamics and learning curves of general task adaptation (SFT and RL), we distinguish our work by specifically characterizing the fundamental properties of LoRA as a dedicated knowledge memory, particularly its storage mechanics and scaling laws in complex, long-context scenarios.

Recent hypernetwork-based approaches generate LoRA adapters directly from textual inputs, such as task descriptions or long contexts, enabling rapid adaptation without per-instance fine-tuning (Charakorn et al., 2025; 2026; Liu et al., 2026). While these works further support LoRA as a compact parametric interface for internalized context or knowledge, they primarily study amortized adapter generation.

It is worth noting that while architectures like Dou et al. (2024), Feng et al. (2024), and Wu et al. (2024) employ learned routing mechanisms similar to Mixture-of-Experts, we exclude such approaches from our analysis for two reasons. First, learned routing often entangles knowledge distribution, making it difficult to trace which module stores specific facts, thereby compromising the modularity required for precise knowledge management. Second, retraining a routing network at test time is computationally impractical for the continuous, plug-and-play learning settings we target.

**Multi-LoRA and Parameter-Space Interpolation.** Another emerging line of work investigates the combination of multiple LoRA modules. Early explorations have shown that interpolating LoRA parameter spaces can compose multiple specialized capabilities (Huang et al., 2023; Feng et al., 2024; Dou et al., 2024; Prabhakar et al., 2025). Recent scalability efforts, such as LAG (Fleshman & Van Durme, 2025c), focus on the routing challenge, proposing efficient retrieval mechanisms to select appropriate modules from libraries containing thousands of LoRAs. Concurrently, PRAG (Su et al., 2025) and DyPRAG (Tan et al., 2025) introduce Parametric RAG frameworks, where document-specific LoRAs are pre-trained

or dynamically generated, and then merged at inference using techniques like TIES-Merging. While these frameworks primarily focus on system-level architectures utilizing short Wikipedia contexts, our work complements them by offering a foundational analysis of LoRA as a long-context memory. We rigorously investigate intrinsic storage properties, including capacity limits, optimal data formatting, and merging scalability.

**Synthetic Data.** Recent work (Lampinen et al., 2025; Park et al., 2025; Lin et al., 2025; Yang et al., 2025b; Zhang et al., 2024b) highlights the importance of data quality and structure for fine-tuning LLMs. Beyond directly training on raw data, studies show that transforming source material into diverse synthetic formats, such as QA pairs, summaries, or rewrites, can improve generalization and robustness. These findings suggest that synthetic augmentation serves as an effective strategy for building denser, more task-aligned knowledge representations.

## B. PhoneBook Benchmark

The PhoneBook benchmark was created to provide a controlled environment for evaluating a model's ability to memorize and recall novel, symbolic associations. This synthetic dataset is designed to be entirely disconnected from the knowledge embedded in the base model during pretraining, thereby isolating the model's capacity for new learning. Its two primary design goals are: (1) to test the pure memorization of arbitrary key-value pairs (fictional name-to-phone number mappings), and (2) to be programmatically scalable, allowing for precise control over the volume of knowledge used in our capacity and efficiency analyses (Section 4).

### Data Generation and Composition

The foundation of the benchmark is a source file, `synthetic_phonebook.csv`, containing a large set of unique, fictional name-phone number pairs. These pairs were generated programmatically to ensure that no real-world entities were included, thus preventing any potential knowledge leakage from the model's pretraining data. Fictional names were synthesized by combining common first and last names, and phone numbers were generated in a standard North American format (`XXX-XXX-XXXX`).

From this source file, we generate training data by converting each name-number pair into a structured Question-Answering (QA) format. This approach frames the information as a natural language question and a direct answer, providing a clear and explicit learning signal for the model. An example of the QA format is shown below.

```
Question: What is the phone number of Jane Doe? Answer: 123-456-7890
```

### Scalable Slicing by Token Count

A key feature of the PhoneBook benchmark is its generation of multiple dataset slices with varying token counts. This allows us to precisely control the amount of information a LoRA module is trained on.

For each target size (e.g., 4K tokens), we iterate through the master source file in a fixed order. Each name-number pair is formatted into the QA format and tokenized using the `Qwen/Qwen3-8B` tokenizer. The formatted entry is added to the dataset slice, and its token count is added to a running total. This process continues until the total token count first exceeds the target size. This deterministic process ensures that smaller datasets are always perfect subsets of larger ones, providing a controlled and consistent methodology for studying the scaling properties of LoRA memory.

### Evaluation Protocol

Performance on the PhoneBook benchmark is measured using a strict **Exact Match (EM)** score. For evaluation, the model is presented with a question asking for the phone number of a specific name included in the training data. The model's generated output is then compared to the ground-truth phone number. A prediction is considered correct only if it is an exact, character-for-character match to the target answer. This strict criterion ensures that we are measuring precise recall, free from any partial credit or semantic ambiguity.

```
Question: "What is the phone number of John Smith?"

Ground Truth Answer: "987-654-3210"

# Correct Prediction (EM Score = 1)
Model Output: "987-654-3210"

# Incorrect Prediction (EM Score = 0)
Model Output: "The phone number for Jane Smith is (987) 654-3210."

# Incorrect Prediction (EM Score = 0)
Model Output: "(987) 654-3210"
```

## C. PaperQA Benchmark

The PaperQA benchmark is designed to rigorously evaluate a model's ability to internalize and reason over novel, complex information. The construction of this benchmark is guided by three core principles: ensuring the novelty of the knowledge, generating a comprehensive set of evaluation questions, and establishing a sophisticated evaluation protocol.

**Source Material and Knowledge Novelty.** To ensure the novelty of the knowledge, we selected source materials from recent top-tier academic conferences. Specifically, we chose five oral or spotlight papers from each of NeurIPS 2024, ICLR 2025, and ICML 2025. We extracted the introduction section from each paper to serve as the knowledge source. Crucially, we verified that the base model had no prior knowledge of these contents before the experiments. By using recent academic papers, we can assess the model's ability to update its factual knowledge and apply its understanding of complex information, allowing for a more precise evaluation of its learning capabilities.

**Comprehensive Question Generation.** To ensure a comprehensive evaluation, we generated with Gemini 2.5 Pro (Comanici et al., 2025), a set of questions using for each document that spans three distinct cognitive levels: (1) Key Information Recall, which tests the retrieval of specific facts; (2) Contextual Comprehension, which assesses the understanding of information in its surrounding context; and (3) Logical Structure Inference, which evaluates the ability to deduce the underlying logical flow and relationships within the text. For each of the 15 papers, we created 10 questions for each cognitive level, resulting in a total of 30 questions per paper. This hierarchical question design enables a multifaceted assessment of knowledge comprehension at various depths.

**Dataset Construction and Evaluation Protocol.** Through this process, we constructed a dataset consisting of 450 question-answer pairs for the 15 new academic papers. To evaluate the model's responses, we employed an LLM-as-a-judge approach, utilizing GPT-4.1 to score the responses with a detailed rubric on a 0-10 scale. This method provides a sophisticated metric that measures the degree of knowledge internalization, moving beyond simple correctness to offer a nuanced assessment of the model's understanding.

**Supplementary Metrics and Robustness Analysis.** While we primarily rely on the LLM judge for its semantic evaluation capabilities, we also report BLEU and ROUGE-L scores in the Appendix to demonstrate the robustness of our performance trends. It is crucial to note that these n-gram-based metrics are not strictly suitable for the open-ended nature of PaperQA, which prioritizes high-level semantic accuracy over lexical overlap. Consequently, absolute scores for these metrics are notably low (e.g., BLEU $\approx 0.1$, ROUGE-L $\approx 0.3$) because the models often generate concise, correct answers that differ structurally from verbose gold references.

As illustrated in the example below, the model provided a factually correct answer, which received a high score from the LLM judge. However, due to the brevity compared to the gold reference, n-gram metrics yielded negligible scores. Despite this discrepancy in absolute values, we observe that the relative performance rankings across different models remain consistent between the LLM judge and traditional metrics.

**Example: Discrepancy between LLM Judge and N-gram Metrics**
**# Case 1**
**Question:** What is the name of the diffusion-based planner cited as an example from Janner et al. (2022)?
**Reference Answer:** The text mentions a diffusion-based planner named Diffuser.

**Predicted Answer:** Diffuser.
**Evaluation Scores:**
- BLEU: $5 \times 10^{-5}$
- ROUGE-L: 0.20
- LLM Judge: 10

---

# Case 2

**Question:** According to the paper, why is it simpler to acquire fine-tuning data for object or attribute changes compared to other edits?
**Reference Answer:** It is simpler because inpainting setups can directly leverage the strong object and attribute abilities of text-to-image models for paired-image data generation.
**Predicted Answer:** It is simpler to acquire fine-tuning data for object or attribute changes because these types of edits have more available paired-image data from text-to-image models.
**Evaluation Scores:**
- BLEU: 0.17
- ROUGE-L: 0.38
- LLM Judge: 8

## C.1. Papers in PaperQA

**ICML 2025**
1. Monte Carlo Tree Diffusion for System 2 Planning
   https://arxiv.org/abs/2502.07202

2. An Analysis for Reasoning Bias of Language Models with Small Initialization
   https://arxiv.org/abs/2502.04375

3. Training Dynamics of In-Context Learning in Linear Attention
   https://arxiv.org/abs/2501.16265

4. Inductive Moment Matching
   https://arxiv.org/abs/2503.07565

5. Statistical Test for Feature Selection Pipelines by Selective Inference
   https://arxiv.org/abs/2406.18902

**NeurIPS 2024**
6. Human Expertise in Algorithmic Prediction
   https://arxiv.org/abs/2402.00793

7. Enhancing Preference-based Linear Bandits via Human Response Time
   https://arxiv.org/abs/2409.05798

8. Rho-1: Not All Tokens Are What You Need
   https://arxiv.org/abs/2404.07965

9. Learning Action and Reasoning-Centric Image Editing from Videos and Simulations
   https://arxiv.org/abs/2407.03471

10. The Value of Reward Lookahead in Reinforcement Learning
    https://arxiv.org/abs/2403.11637

**ICLR 2025**
11. Artificial Kuramoto Oscillatory Neurons
    https://arxiv.org/abs/2410.13821

12. Exploring the Loss Landscape of Regularized Neural Networks via Convex Duality
    https://arxiv.org/abs/2411.07729

13. In Search of Forgotten Domain Generalization
    https://arxiv.org/abs/2410.08258

14. Programming Refusal with Conditional Activation Steering
    https://arxiv.org/abs/2409.05907

15. Efficient and Accurate Explanation Estimation with Distribution Compression
    https://arxiv.org/abs/2406.18334

## C.2. Generation Prompts

```
You are an expert academic assistant tasked with creating a high-quality question-answering dataset from a
    research paper's introduction. Your goal is to generate 30 question-and-answer pairs based exclusively on
    the provided text.

Instructions and Rules:
Source Grounding: All questions and answers MUST be derived solely from the provided introduction text. Do not
    use any external knowledge or make assumptions beyond what is written.

Question Hierarchy: You must create questions across three distinct levels of understanding, as defined below.

Quantity: Generate exactly 30 pairs in total: 10 for Level 1, 10 for Level 2, and 10 for Level 3.

Output Format: The output must be a single, valid JSON array of objects. Do not include any explanatory text,
    comments, or markdown formatting before or after the JSON code block.

Question Level Definitions:
Level 1: Key Information Recall (10 Questions)

Objective: To test the recall of specific, explicitly stated facts, proper nouns, terminology, and figures from
    the text.

Question Type: "What is...?", "What are the names of...?", "Which X was mentioned...?"

Level 2: Contextual Comprehension (10 Questions)

Objective: To test the understanding of relationships between concepts, such as cause-and-effect, problem-
    solution, comparisons, and the function of a component.

Question Type: "Why does...?", "What is the effect of A on B?", "How does X work?", "What is the difference
    between A and B?"

Level 3: Logical Structure Inference (10 Questions)

Objective: To test the understanding of the overall logical flow of the text, including identifying the core
    problem, the research gap, the proposed solution, and the main contribution.

Question Type: "What is the core problem the authors aim to solve?", "What research gap does this paper intend to
     fill?", "What is the main advantage of the proposed method?", "Summarize the key contribution of this work
     in relation to prior limitations."

Desired JSON Output Format:
The final output MUST be a JSON array containing 30 objects. Each object must have three keys: level (integer: 1,
     2, or 3), question (string), and answer (string).

Example:

[
  {
    "level": 1,
    "question": "What is the full name of the algorithm the authors integrate their estimator into?",
    "answer": "The authors integrated their estimator into the Generalized Successive Elimination algorithm."
  },
  {
    "level": 2,
    "question": "According to the text, what is the inverse relationship between response time and preference
        strength?",
    "answer": "The text states that users who strongly prefer to skip a product tend to do so quickly, while
        longer response times can indicate weaker preferences."
  },
```

```
  {
    "level": 3,
    "question": "What is the core reason complex psychological models are impractical for real-time systems, and
        how does this paper's proposal address it?",
    "answer": "They are impractical because they rely on computationally intensive methods like hierarchical
        Bayesian inference and MLE. This paper addresses it by proposing a computationally efficient method that
            frames utility estimation as a linear regression problem."
  }
]

Introduction Text to Analyze:
[INSERT INTRODUCTION TEXT HERE]

Now, generate the 30 Q&A pairs in the specified JSON format based on the text provided above.
```

## C.3. Judging Prompts

```
You are an impartial AI assistant acting as an expert judge. Your task is to evaluate a candidate's answer to a
    question about a technical document. Compare the candidate's answer against the gold standard answer.

[EVALUATION CRITERIA]
1.  **Factual Alignment**: Does the candidate answer state the same facts as the gold answer? It must not
    contradict the gold answer.
2.  **Completeness**: Does the candidate answer include all the key information and nuances present in the gold
    answer?
3.  **Relevance**: Is the answer focused and on-topic? It must not contain irrelevant or hallucinatory
    information.

[SCORING RUBRIC (0-10 SCALE)]
- **10**: Perfect. The candidate answer is factually identical to the gold answer, complete, and contains no
    extraneous information.
- **7-9**: Mostly Correct. The answer is factually correct but might omit a minor detail or be slightly verbose.
    The core information is present and accurate.
- **4-6**: Partially Correct. The answer has the right general idea but contains a significant factual error, a
    major omission, or irrelevant information.
- **1-3**: Incorrect. The answer is on-topic but factually wrong.
- **0**: Completely Incorrect. The answer is nonsensical, irrelevant, or fails to address the question.

[TASK]
Evaluate the [CANDIDATE ANSWER] based on the criteria above and its alignment with the [GOLD ANSWER]. Provide
    your output in a single JSON object with two keys: "score" (an integer from 0-10) and "rationale" (a brief,
    one-sentence explanation for your score).

[QUESTION]
{question}

[GOLD ANSWER]
{gold_answer}

[CANDIDATE ANSWER]
{predicted_answer}
```

## D. Experimental Settings

To facilitate reproducibility and provide a clear overview of our extensive experimental evaluation ($Q1$ through $Q15$), we summarize the master configuration for all experiments in this section. Table 3 maps each research question to its specific experimental setup, including the dataset, base model, and key variable parameters.

*Table 3.* Master experiment configuration. Each questions are separated by thin lines, while major sections are divided by full-width lines.

| Section | ID | Topic (RQ) | Dataset | Base Model(s) | Key Variables & Settings |
|---|---|---|---|---|---|
| **4** | Q1 | Scalability w/ Rank | CF | Llama-3.1-8B Qwen3-8B | **Rank:** $r \in \{2, 4, \dots, 1024\}$ |
| | Q2 | Capacity Limit | PB, CF | Llama-3.1-8B Qwen3-8B | **Data:** 1K $\rightarrow$ 20K tokens (step 1K) **Rank:** Fixed per run ($r \in \{2..1024\}$) |
| | Q3 | Param. Efficiency | PB, CF | Llama-3.1-8B Qwen3-8B | **Metric:** Efficiency $= T_{max}/N_{params}$ **Threshold:** $\tau \in \{0.8, 0.9, \dots\}$ |
| **5** | Q4 | Synthetic Formats | PaperQA | Llama-3.1-8B Qwen3-8B | **Format:** Original, QA{10, 20, 40}, Summary{2, 4, 8}, Rewrite{1, 2, 4} |
| | Q5 | Data Mixing | PaperQA | Llama-3.1-8B Qwen3-8B | **Format:** Combination of Original, Summary8, Rewrite4 and QA40 |
| | Q6 | Base Model Scale | PaperQA | Qwen3 Family | **Size:** 0.6B, 1.7B, 4B, 8B, 14B **Params:** Specific learning rate, training step per model |
| | Q7 | Generator Quality | PaperQA | Llama-3.1-8B Qwen3-8B | **Generator:** base model, GPT-4.1 **Data:** QA20 |
| **6** | Q8 | Multi-LoRA PoC | PB (64K) | Llama-3.1-8B Qwen3-8B | **Setup:** Single ($r = 32$, 64K) vs. Multi ($r = 4$, 8 modules) **Routing:** Oracle |
| | Q9 | Routing Error | PaperQA | Llama-3.1-8B Qwen3-8B | **Routing:** Oracle vs. Embedding (BGE) **Baseline:** Single LoRA trained on all document |
| | Q10 | Merging Strategy | PaperQA | Llama-3.1-8B Qwen3-8B | **Merge:** Linear, Cat, TIES, DARE **Routing:** Top-3 Embedding-based Retrieval |
| | Q11 | Number of Merges | PaperQA | Llama-3.1-8B Qwen3-8B | **Merge:** TIES **Routing:** $1 \rightarrow 5$ ground-truth set modules |
| **7** | Q12 | Long Context QA | NQA QuALITY | Llama (1B, 8B) Qwen (1.7B, 8B) | **Chunking:** 2048 tokens ,200 overlap (NQA) / 768 tokens, 77 overlap (QuALITY) **Rank:** Multi ($r = 4$), Single ($r = 16$) |
| | Q13 | Hybrid Context | NQA QuALITY | Llama (1B, 8B) Qwen (1.7B, 8B) | **Hybrid:** LoRA + ICL / LoRA + RAG **Comparison:** Standalone ICL, RAG |
| | Q14 | Merging Continuity | NQA QuALITY | Llama (1B, 8B) Qwen (1.7B, 8B) | **Merge:** Comparing Top-1 vs. Top-3 TIES Merging |
| | Q15 | Latency / Cost | NQA | Llama-3.1-8B | **Mode:** Preloaded vs. Dynamic Loading **Metric:** End-to-end wall-clock time |

**Note on Hyperparameters:**

- **PB/CF (Q1–Q3, Q8):** Training is conducted for 1,500 steps with a batch size of 8. We set the scaling factor $\alpha$ equal to the rank ($\alpha = r$).

- **PaperQA (Q4–Q7, Q9–Q11):** Training is conducted for 1,000 steps with a learning rate of $5 \times 10^{-5}$ and a batch size of 8. We adopt a default rank of $r = 16$ with $\alpha = r$.

- **NQA/QuALITY (Q12–Q14):** We use a batch size of 32 and a learning rate of $5 \times 10^{-4}$. For multi-LoRA, we use $r = 4, \alpha = 8$ trained for 150 steps; for single LoRA, we use $r = 16, \alpha = 32$ trained for 250 steps.

- **Hardware:** All experiments were conducted on NVIDIA RTX PRO 6000 Blackwell GPUs.

# E. Additional Experimental Results

We report additional experiments that were omitted from the main body due to space constraints. These experiments examine (i) **where** to apply Knowledge LoRA (layer/module ablations), and (ii) whether **LoRA variants** provide meaningful gains over standard LoRA in our memory setting.

## E.1. Layer and Module Ablations

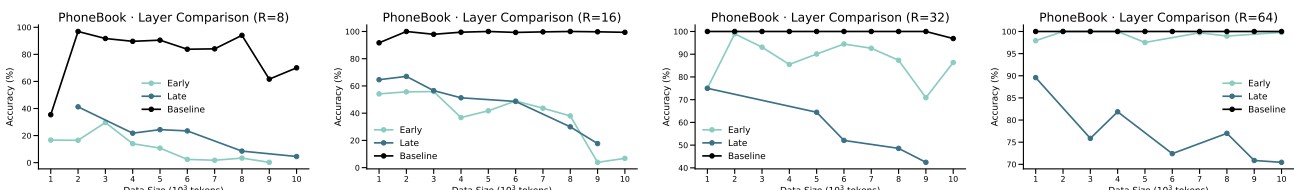

*Figure 9.* PhoneBook layer ablation (Early vs. Late) under ranks $r \in \{8, 16, 32, 64\}$.

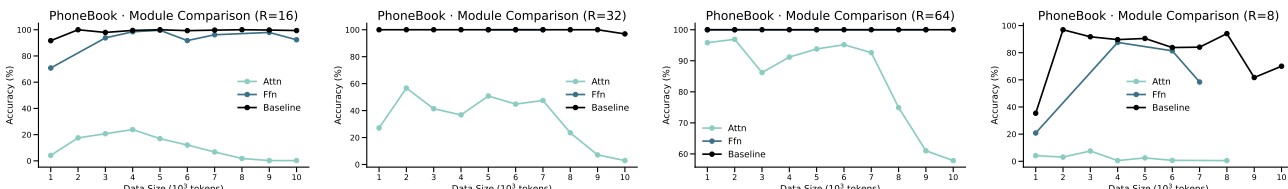

*Figure 10.* PhoneBook module ablation (Attn vs. FFN) under ranks $r \in \{8, 16, 32, 64\}$.

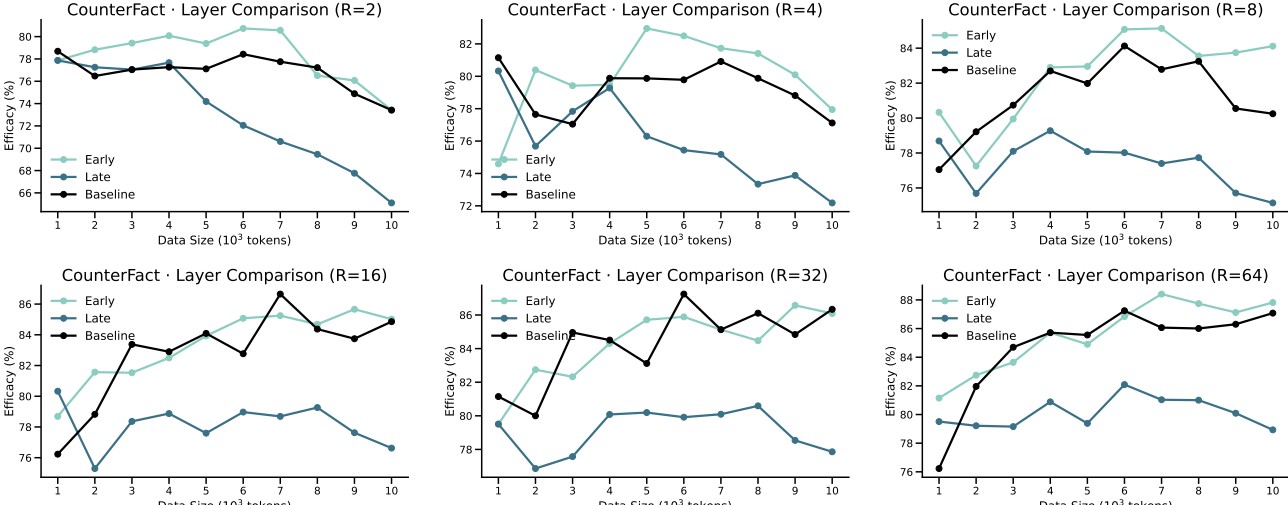

*Figure 11.* CounterFact layer ablation (Early vs. Late) under ranks $r \in \{2, 4, 8, 16, 32, 64\}$.

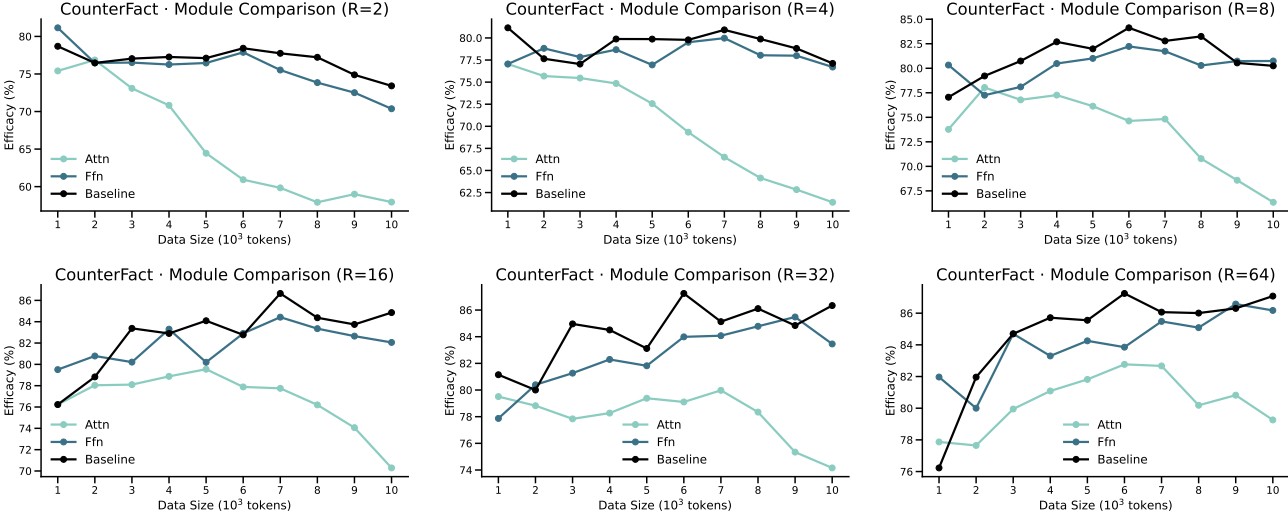

*Figure 12.* CounterFact module ablation (Attn vs. FFN) under ranks $r \in \{2, 4, 8, 16, 32, 64\}$.

**Motivation.** In Section 4, we discuss a rank-dependent saturation effect, where performance collapses once the knowledge load exceeds a LoRA module's capacity. In practice, LoRA does not need to be applied to every layer and every sub-module: one may target only a subset of layers (e.g., early vs. late) or only certain sub-modules (e.g., attention vs. feed-forward). Prior notes on LoRA suggest that FFN-targeting can sometimes outperform attention-targeting (Fomenko et al., 2024), while LoRA without regret argues that including FFN across layers is generally beneficial (Schulman & Lab, 2025). We therefore test whether selective placement alleviates saturation or improves parameter efficiency in our knowledge-memorization regime.

**Setup.** We use Llama-3.1-8B as the base model and follow the same hyperparameters with Q1, varying the knowledge load for PhoneBook (PB) and CounterFact (CF). Llama has 32 transformer layers; for layer ablation, we define **Early** = first 16 layers and **Late** = last 16 layers. For module ablation, we apply LoRA only to (i) **attention** projections, or (ii) **FFN** linear layers. We compare against the baseline configuration that applies LoRA to all layers/modules (our default in the main paper).

**Layer ablation.** Across CF (and PB in low-rank regimes), applying LoRA to Early layers generally yields stronger performance than Late, and Late layers tend to saturate earlier as the knowledge load increases (Figures 9 and 11). This trend becomes clearer at higher ranks; in PB with small ranks, Early/Late can appear similar, but in CF and PB low-rank settings, Early is more consistently advantageous. Notably, in CF, we occasionally observe Early outperforming the all-layer baseline, suggesting a slightly different behavior than the "apply everywhere" guidance of (Schulman & Lab, 2025) in this specific knowledge-memory setup.

**Module ablation.** In all tested settings, FFN-only LoRA consistently outperforms attn-only LoRA (Figures 10 and 12). This aligns with observations in prior guidance (Fomenko et al., 2024; Schulman & Lab, 2025) and with claims that FFN plays a central role in storing factual knowledge, motivating FFN-targeted edits (Meng et al., 2023).

Although selective placement can improve performance, we found it to be more sensitive to hyperparameters and sometimes unstable: in PB, some low-rank selective configurations failed to train reliably. For robustness and reproducibility, the main paper therefore adopts the most common and stable practice: applying LoRA broadly across layers/modules. A deeper study of these training dynamics for knowledge LoRA is a promising direction for future work.

### E.2. LoRA Variants: DoRA and PiSSA

**Motivation.** Beyond placement, one may ask whether replacing standard LoRA with a lightweight variant yields better knowledge memory. Among many variants, we choose **DoRA** (Liu et al., 2024) and **PiSSA** (Meng et al., 2024) as they (i) are widely-used, (ii) are readily supported in common PEFT toolchains, and (iii) remain close in spirit to LoRA (i.e., do not require major system changes). We evaluate them on PaperQA and NarrativeQA.

**Setup.** For PaperQA, we mirror the single LoRA training setup used in Section 5 and swap the adapter type (LoRA / DoRA / PiSSA). For NQA, we repeat the core settings from Section 7 and compare single, multi-LoRA Top-3, and multi-LoRA Top-3 + ICL for Llama-3.1-8B and Qwen3-8B.

**Results.** The overall trends are not consistent across models and settings. On PaperQA, PiSSA improves performance for Llama in our runs, while for Qwen, it is the least effective among the three; DoRA is competitive, but the gap is small (Figure 13). On NQA, we do not observe a systematic benefit from switching variants, except that on Qwen the variants can improve the multi-LoRA Top-3 + ICL configuration (Table 4). We therefore conclude that simply swapping to DoRA/PiSSA does not reliably yield large gains in our knowledge-memory setting.

# F. Details of Q1. Is Memory Capacity Scalable with LoRA Rank?

## Motivation

The viability of employing Low-Rank Adaptation (LoRA) as a knowledge memory module for Large Language Models (LLMs) hinges on a fundamental question: Is its capacity scalable? If the amount of information LoRA can store is severely limited or cannot be reliably controlled, its utility as a memory architecture is questionable. For researchers and practitioners, the LoRA rank is the most direct and accessible hyperparameter for modulating model capacity. However, a systematic analysis of the explicit relationship between LoRA rank and memory capacity has been notably absent in prior work.

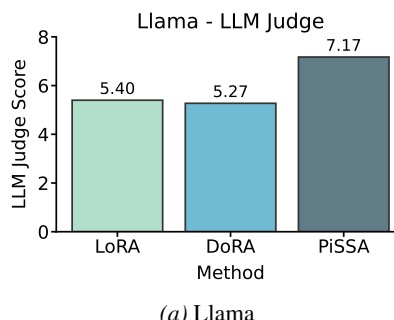

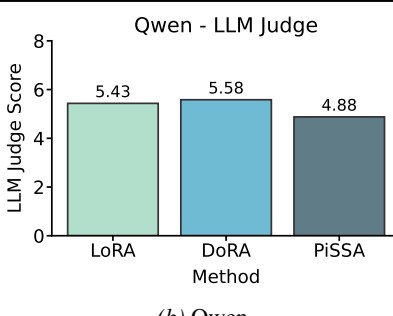

*(a)* Llama

*(b)* Qwen

*Figure 13.* LoRA variant comparison. We compare standard LoRA, DoRA, and PiSSA under the same experimental protocol, using PaperQA. **(left)** Llama results. **(right)** Qwen results.

*Table 4.* **NarrativeQA: LoRA variant comparison.** We report ROUGE-L for standard LoRA, DoRA, and PiSSA on representative NQA configurations (Single, Multi-LoRA Top-3, and Multi-LoRA Top-3 + ICL).

| Model | Method | Single | Multi (Top3) | Multi + ICL |
|---|---|---|---|---|
| | LoRA | **27.05** | 22.42 | **38.78** |
| Llama-3.1-8B | DoRA | 25.04 | 22.96 | 38.41 |
| | PiSSA | 8.56 | **23.38** | 38.34 |
| | LoRA | 25.78 | **21.15** | 37.23 |
| Qwen3-8B | DoRA | **27.48** | 20.52 | **40.82** |
| | PiSSA | 24.33 | 20.80 | 40.51 |

Understanding this relationship is crucial for addressing the practical question: *What rank is required to store a given amount of knowledge?* Answering this question will establish foundational guidelines for the principled design and efficient resource allocation of LoRA-based memory modules, moving beyond ad-hoc selection of the rank parameter.

**Experimental Setup**

We use CF dataset with a 8K, 10K, 12K, 14K token length. For other experimental setups, we followed Appendix D.

**Experimental Results**

Across both the PhoneBook and CounterFact datasets, we observe a consistent and positive correlation between the LoRA rank and the model's performance. As illustrated in Figure 1, performance on the CounterFact dataset shows a distinct monotonic increase as the rank is scaled from 2 to 1024. This trend demonstrates that allocating more parameters to the LoRA module—by increasing its rank—directly enhances its capacity to reliably internalize a larger or more complex body of information.

However, we also observe a pattern of diminishing returns. The marginal performance gain for each incremental increase in rank tends to decrease, especially at higher rank values. This suggests that while capacity consistently grows with rank, the parameter efficiency for storing new knowledge diminishes. This trade-off between absolute capacity and efficiency is a critical consideration for the practical application of LoRA as a memory module.

## G. Details of Q2. Is There a Finite Capacity Limit? How Is It Determined?

**Motivation**

While our initial findings confirm the scalability of LoRA's memory capacity with rank, this scalability cannot be infinite. For LoRA to be reliably integrated into practical systems as a memory component, it is imperative to move beyond observing scalability and instead identify its finite capacity limits. Understanding where these limits lie and how they are determined by the model's architecture is not merely a theoretical question. It has direct and critical implications for system design, informing how to provision resources and predict model behavior when faced with varying knowledge loads. *At what point does a LoRA module of a given rank become saturated, and what governs this saturation point?*

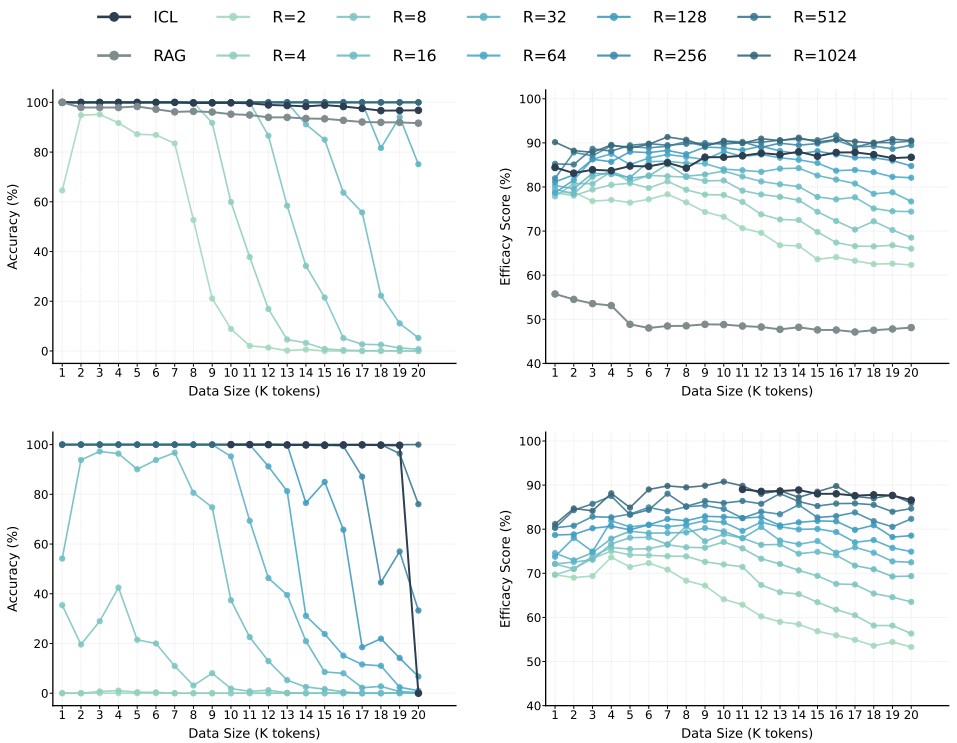

*Figure 14.* Full results for memory capacity experiment on Llama (top row) and Qwen (bottom row). **(Left)** Performance on PhoneBook for various ranks as data length increases. **(Right)** Performance on CounterFact for various ranks as data size increases.

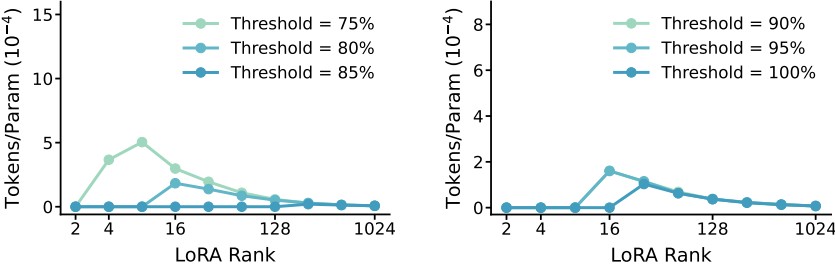

*Figure 15.* Efficiency for the two datasets (Qwen). **(Left)** CounterFact results. **(Right)** PhoneBook results.

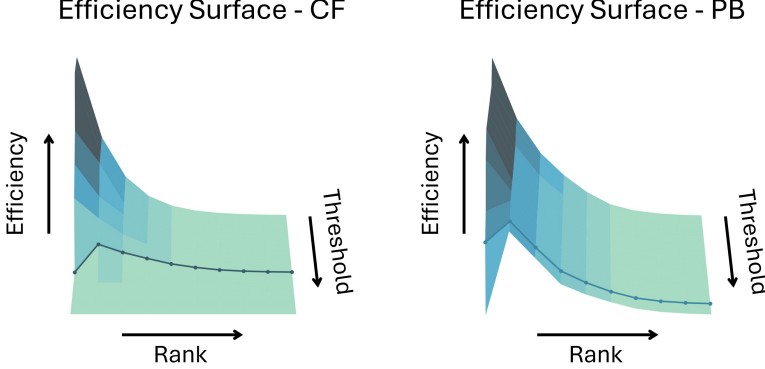

*Figure 16.* Surface generated by rank, threshold and efficiency measured on Llama. **(Left)** CounterFact. **(Right)** PhoneBook.

**Experimental Setup**

All other hyperparameters and experimental conditions adhere to the configurations detailed in the Appendix D. This setup allows us to observe the performance of each rank configuration as it is exposed to an increasing amount of information, thereby revealing its saturation point.

**Experimental Results**

Our experiments reveal that a LoRA module possesses a clear, finite memory capacity, and this limit is fundamentally determined by its rank. We observe a distinct saturation phenomenon across both datasets. For any given rank, performance remains high or increases as the data volume grows, but only up to a certain threshold. Beyond this point, performance degrades sharply, indicating that the module's capacity has been exceeded.

Crucially, this saturation point is a direct function of the rank. As depicted in Figure 14, higher-rank LoRA modules are capable of internalizing a significantly larger volume of knowledge before their performance collapses. This explicitly shows that a larger rank effectively raises the capacity ceiling.

These results yield a critical guideline for practitioners: the total volume of knowledge to be stored must be carefully matched with the allocated LoRA rank. Attempting to inject a large corpus of information into a low-rank LoRA module will inevitably lead to performance degradation due to this capacity overflow. Therefore, developers must provision a rank sufficient for the target knowledge volume to ensure stable and successful knowledge internalization.

## H. Details of Q3. Is Using The Highest Rank Always The Most Efficient Choice?

**Motivation**

The findings from our previous sections confirm that LoRA's capacity scales with rank, which might suggest a straightforward strategy: use the highest rank available within resource constraints. However, this more is better approach requires rigorous validation from a cost-benefit perspective. This section challenges this naive assumption by investigating whether the benefit of increased storage capacity is always proportional to the cost of a higher rank (i.e., more parameters, longer training times, and a larger memory footprint). If a regime of diminishing returns exists—where doubling the rank fails to double the effective capacity—then indiscriminately increasing the rank is a suboptimal strategy.

To formalize this, we move beyond raw performance and introduce a quantitative metric for efficiency: **Parameter Efficiency**, defined as the amount of information stored per parameter. This metric allows for a fair and objective comparison between LoRA modules of different ranks, enabling us to answer the question of how much knowledge can be stored per unit of cost. By analyzing this, we can identify an optimal design specification, or a sweet spot, for a single LoRA module, revealing the rank at which it is most efficient at compressing and internalizing new information.

**Experimental Setup**

To evaluate parameter efficiency, we systematically vary the LoRA rank and measure its corresponding storage capability. We set the LoRA rank $r$ to span exponentially from 2 to 1024 ($r \in \{2, 4, 8, \ldots, 1024\}$). For each rank, we measure the Parameter Efficiency using the formulation from Equation 1 across various performance thresholds. This allows us to map out the efficiency landscape as a function of rank. For other experimental setups, we followed Appendix D.

**Experimental Results**

Our results provide an unequivocal answer: No, the highest rank is not the most efficient choice. The efficiency measurements, plotted in Figure 3 and Figure 15 (made by slicing Figure 16, in specific threshold) reveal that the parameter efficiency curve is non-monotonic. Instead of increasing with rank, the curve peaks at a specific point and subsequently declines. This indicates that beyond an optimal point, the growth in parameter count outpaces the effective gains in memory capacity, leading to decreased overall efficiency.

Specifically, across both the PhoneBook and CounterFact datasets, we observe peak storage efficiency in the low-rank regime. The most efficient configuration was consistently found at a small rank, such as $r = 4$. This finding strongly suggests that smaller, more compact LoRA modules can be significantly more parameter-efficient for knowledge storage than their larger counterparts.

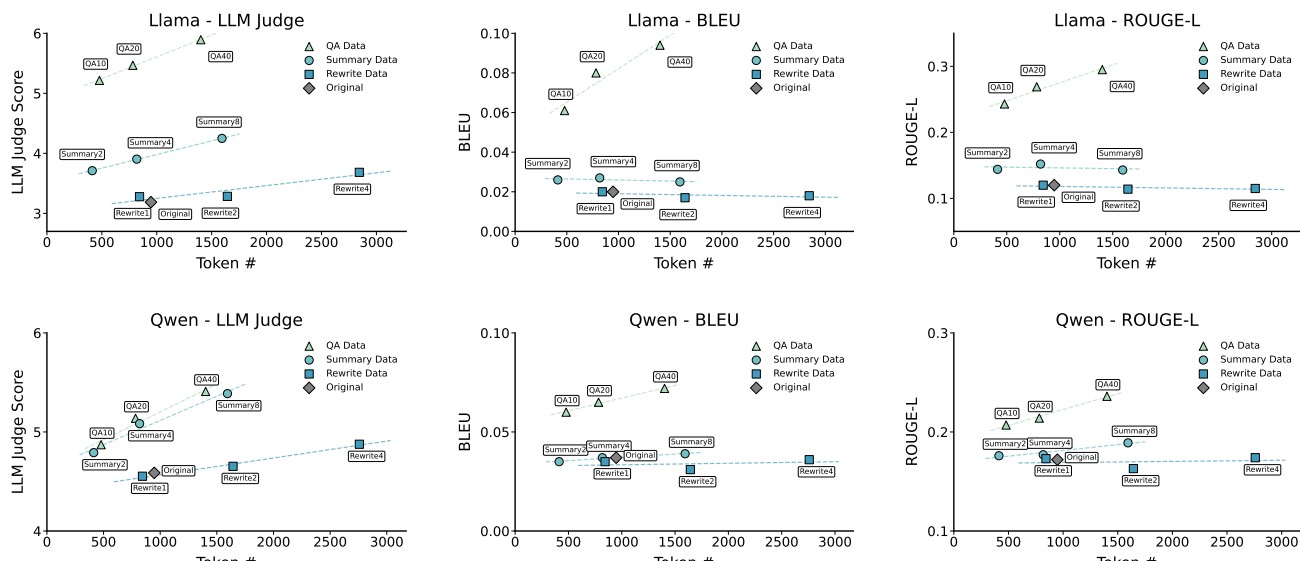

*Figure 17.* Performance scaling with different synthetic data generation methods. **(Top)** Llama performance across LLM judge, BLEU, and ROUGE metrics.**(Bottom)** Qwen performance across LLM judge, BLEU, and ROUGE metrics.

# I. Details of Q4. How Does Synthetic Data Enhance Single LoRA's Knowledge Memorization?

**Motivation**

Our analysis has established that a LoRA module possesses a finite memory capacity. Training such a module on raw text alone presents a challenge, as the model may struggle to discern which information is critical, akin to learning from a textbook without highlighted key concepts or practice questions. This raises a fundamental question: how can we refine raw data into more effective learning signals to maximize the utility of LoRA's limited capacity? This section investigates strategies for transforming source documents into synthetic data for more efficient knowledge internalization.

We explore two primary hypotheses. First, we consider the impact of knowledge density and format. It is plausible that different data structures, such as question-answer (QA) pairs or summaries, may offer more compressed and potent learning signals than unstructured narrative text. Second, we examine the scalability of synthetic data generation. While prior studies have indicated the benefits of synthetic data, it is not well-established whether performance scales monotonically with the quantity of synthetic data derived from a single source. Answering these questions can provide practical guidelines for constructing an optimal data processing pipeline for LoRA-based knowledge memorization.

**Experimental Setup**

We compare three synthetic data generation strategies against a raw text baseline: **(1) Question-Answering (QA)**, **(2) Summary**, and **(3) Rewrite**. To analyze the impact of data scaling, we experimented with varying quantities for each strategy: 10, 20, and 40 pairs for QA; 2, 4, and 8 variations for Summary; and 1, 2, and 4 variations for Rewrite. All synthetic data was generated using GPT-4.1. For other experimental setups, we followed Appendix D.

**Experimental Results**

The results suggest that transforming raw text into structured, synthetic data is a highly effective strategy for enhancing knowledge memorization in LoRA.

As shown in Figure 4 and Figure 17, all three synthetic data formats—QA, Summary, and Rewrite—resulted in markedly better performance than the raw text baseline. This observation supports the hypothesis that synthetic data generation provides a clearer and more potent learning signal for the model.

Among the different formats, those involving information compression appeared to be the most effective. The observed order of performance, from most to least effective, was **QA**, followed by **Summary**, **Rewrite**, and finally the raw text baseline. The QA and Summary formats, which distill key information, outperformed the less structured Rewrite and raw text formats.

The QA format's top performance may be partially attributed to its structural alignment with the evaluation task, which is also QA-based. This suggests that consistency between the training data format and the target task is an important factor.

Finally, while performance tended to improve with a larger quantity of synthetic data for all methods, the efficiency of this improvement varied. The performance gain per token was highest for the QA format, followed by Summary and then Rewrite. This finding implies that the method of synthetic data generation and the quality of the resulting data may be more critical than the sheer volume of data used.

### I.1. Generation Prompts for QA

```
You are an expert question-answer pair generator.
Based on the text provided, generate exactly {num_samples} question-answer pairs.
Your output must be ONLY a single JSON object with a key "items" that contains a list of objects. Do not include
    any text outside the JSON. Each object in "items" must have two keys: "question" and "answer".

[TEXT]
{text}
```

### I.2. Generation Prompts for Summary

```
Based on the text provided, generate exactly {num_samples} distinct summaries of the content.
Your output must be ONLY a single JSON object with a key "items" that contains a list of objects. Do not include
    any text outside the JSON. Each object in "items" must have a single key: "summary".

[TEXT]
{text}
```

### I.3. Generation Prompts for Rewrite

```
Based on the text provided, generate exactly {num_samples} distinct rewrites of the passage.
While keeping entities and key details intact, use different vocabulary and sentence structures.
Ensure that the length of each rewrite is approximately the same as the original text (aim to maintain similar
    word count and overall length).
Your output must be ONLY a single JSON object with a key "items" that contains a list of objects. Do not include
    any text outside the JSON. Each object in "items" must have a single key: "rewrite".

[TEXT]
{document_text}
```

## J. Details of Q5. What Are the Effects of Combining Synthetic Data Formats?

**Motivation**

The previous section identified the Question-Answering (QA) format as a particularly effective single method for synthetic data generation. This leads to a natural follow-up question: can combining different formats yield synergistic effects that surpass the performance of the best single format? Investigating this is essential for exploring new possibilities to move beyond single method optimization and potentially achieve higher performance ceilings.

Furthermore, this inquiry offers a deeper understanding of LoRA's learning mechanisms. Different data formats may encourage the learning of distinct cognitive skills; for example, QA might enhance factual retrieval, summaries could foster high-level conceptual understanding, and rewrites might improve stylistic flexibility. By combining these formats, we can observe whether LoRA can integrate these varied learning signals in a complementary fashion. While the principle of data diversity is often considered beneficial, its specific scaling effects in the context of synthetic data for LoRA-based memorization have not been systematically explored. This study aims to address this by examining how performance responds to an increasing variety of data formats.

**Experimental Setup**

This experiment is a direct continuation of the analysis in the previous section, utilizing the same synthetic datasets. To investigate the impact of data diversity, we employed four specific data formats derived from the source material: Original, QA40, Summary8, and Rewrite4.

We constructed new, mixed-format training datasets by concatenating the text files of these individual components. For instance, the combination labeled 'Summary8 + QA40' was created by merging the dataset containing 8 summaries with the one containing 40 QA pairs. We evaluated various combinations of these formats, as detailed in Table 1, to identify the most effective data configuration. For other experimental setups, we followed Appendix D.

**Experimental Results**

Our results demonstrate that leveraging diverse synthetic data formats consistently enhances performance across different model architectures. As shown in Table 1, the integration of multiple data formats yields a robust improvement over single format baselines. Regardless of the specific architecture, introducing variation in data representation (e.g., combining QA with Summary or Rewrite) consistently led to higher token retention compared to the Original method. This suggests that diversifying the training objective helps the model internalize knowledge more effectively, irrespective of the underlying model capacity or structure.

## K. Details of Q6. How Does the Scale of Base Models Affect LoRA Knowledge Internalization?

**Motivation**

LoRA does not operate in isolation; it functions as an adapter that attaches to a pre-trained base model. Consequently, its final performance is likely influenced not only by its own parameters but also by the intrinsic capabilities of the underlying model. This dependency raises a critical question regarding the interplay between the adapter and the base model: does a larger, more capable base model provide a more fertile foundation for LoRA to effectively integrate and leverage new knowledge?

Understanding this relationship is crucial for assessing the practical deployment and cost-effectiveness of LoRA-based memory systems. The degree to which LoRA's effectiveness depends on the base model's scale helps to clarify the performance trade-offs involved in system design. If substantial performance gains are only achievable with large-scale models, the applicability of this method might be better suited for high-resource environments. Conversely, if LoRA can significantly augment the capabilities of smaller models, it presents a viable strategy for achieving competent performance within more constrained resource budgets.

**Experimental Setup**

To isolate the effect of model scale on LoRA's knowledge internalization, we selected various models from the Qwen3 family. The specific models used were Qwen3-0.6B, Qwen3-1.7B, Qwen3-4B, Qwen3-8B, and Qwen3-14B . This approach allowed us to directly attribute performance differences to the inherent capacity of the base model. To ensure that the base model's scale was the sole variable. To find optimal hyperparameters for each model, we swept over learning rates ($\eta \in \{1e-5, 5e-5, 1e-4, 5e-4\}$) and training steps ($S \in \{250, 500, 1000, 2000\}$) to identify the optimal setup for each Qwen variant. The selected hyperparameters are listed in Table 5, while other experimental configurations follow the details in Appendix D.

*Table 5.* Optimal hyperparameters identified via grid search for each Qwen3 model variant.

| Model Variant | Learning Rate ($\eta$) | Training Steps ($S$) |
|---|---|---|
| Qwen3-0.6B | $5 \times 10^{-4}$ | 250 |
| Qwen3-1.7B | $5 \times 10^{-4}$ | 1000 |
| Qwen3-4B | $1 \times 10^{-4}$ | 500 |
| Qwen3-8B | $5 \times 10^{-4}$ | 1000 |
| Qwen3-14B | $5 \times 10^{-4}$ | 250 |

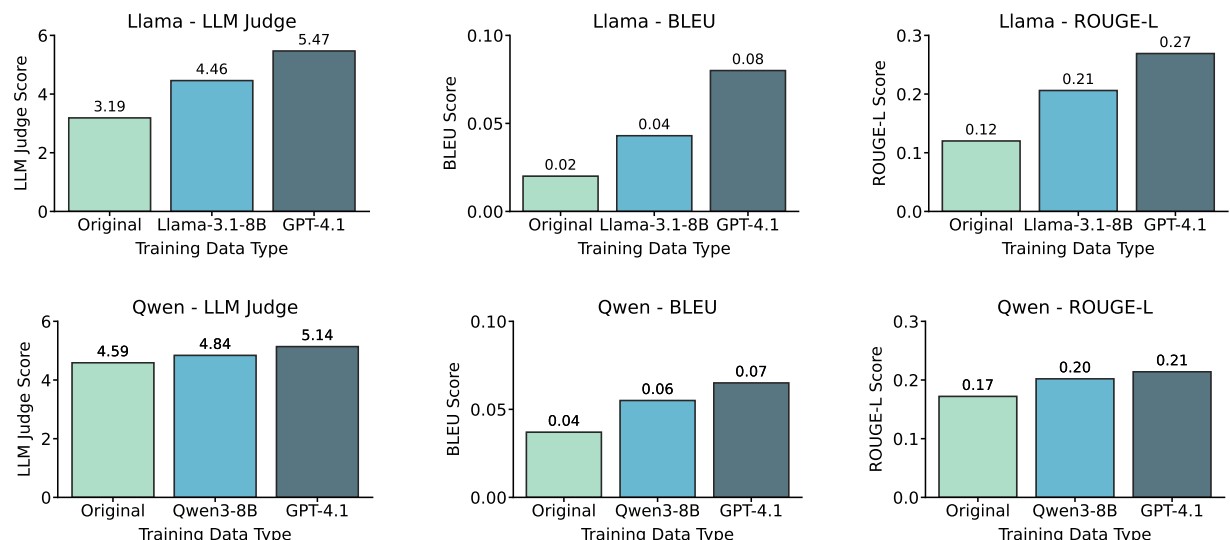

*Figure 18.* Performance comparison of synthetic data generator. **(Top)** Results for Llama across LLM judge, BLEU, and ROUGE metrics. **(Bottom)** Results for Qwen across LLM judge, BLEU, and ROUGE metrics.

## Experimental Results

Our results indicate a general trend where performance improves as the size of the base model increases. As depicted in Figure 5 (left), the PaperQA score consistently rises with the parameter count, from the 0.6B to the 14B model. This suggests that larger models may indeed provide a more advantageous foundation for internalizing and applying new knowledge.

However, this performance enhancement is not directly proportional to the model's scale and exhibits a notable non-linear pattern. We observed significant performance gains when scaling from 0.6B to 1.7B and again from 8B to 14B. In contrast, the performance improvement was relatively marginal across the intermediate-sized models, specifically from 1.7B to 8B.

This non-linear relationship may offer insights into LoRA's operational mechanics. Unlike methods such as in-context learning, which heavily leverage the base model's intrinsic reasoning abilities, LoRA appears to focus more on directly storing new knowledge within its own added parameters. This could imply a relatively lower dependency on the base model's scale compared to other methods. Furthermore, it is noteworthy that even a relatively small model, such as the 1.7B variant, achieved a substantial level of knowledge processing capability when augmented with LoRA.

These findings have important implications for practical applications. The non-linear scaling suggests that a cost-benefit analysis is crucial when selecting a base model. For instance, in an operational range corresponding to our observed plateau region (1.7B to 8B), upgrading to a larger base model may yield only minimal returns. In such cases, focusing on improving data quality or tuning LoRA's rank could be a more cost-effective optimization strategy. While using the largest available model can generally be expected to yield higher performance, our results suggest that combining a well-optimized LoRA with a mid-sized model could be a more rational and efficient alternative for achieving specific performance targets.

## L. Details of Q7. Does Synthetic Data Generator Quality Impact LoRA Performance?

**Motivation**

The quality of training data is a foundational factor that often determines final model performance in machine learning. This section investigates the hypothesis that for synthetic data, the quality of the data-generating model directly influences the quality of the resulting training data and, consequently, the performance of the trained LoRA module.

This inquiry is also motivated by a practical need for a clear cost-benefit analysis. High-performance models such as GPT-4 incur high costs for data generation, whereas open-source models like Llama offer a more cost-effective alternative. By quantifying the performance difference, this study aims to provide developers with practical guidance: does the investment in a superior, high-cost generator yield a proportional performance benefit, or can more accessible models produce data of sufficient quality?

**Experimental Setup**

To analyze the impact of the synthetic data generation model on LoRA's performance, we compared two distinct models: the high-performance, API-based GPT-4.1 and the accessible, local base model (Llama-3.1-8B and Qwen3-8B). For a fair and direct comparison, we standardized the amount of synthetic data to 20 QA pairs per document for both generation models. This adjustment was necessary because preliminary experiments showed that the local models had difficulty reliably generating the larger set of 40 QA pairs used in other experiments. For other experimental setups, we followed Appendix D.

**Experimental Results**

The results indicate that the quality of the data-generating model appears to have a substantial impact on the final performance of the LoRA module. As shown in Figure 5 (right) and Figure 18, there is a clear performance disparity between the LoRAs trained on data from the two different generators. This gap seems to directly reflect the known capability difference between the generator models themselves. This performance difference can plausibly be attributed to the quality of the generated data; it is likely that the more advanced model produced training examples that were more accurate, logically coherent, and comprehensive.

This finding has a significant implication for the use of LoRA as a knowledge module. The performance of a knowledge-infused LoRA is not only dependent on its own architecture or training process but is also deeply tied to the quality of its knowledge source. The process can be conceptualized as a form of knowledge distillation, where the knowledge contained within the large generator model is transferred to the more compact LoRA module. Consequently, the depth and breadth of knowledge that the generator model can comprehend and articulate may act as a practical upper bound on the knowledge that the LoRA module can effectively learn and represent.

## M. Details of Q8. How Can We Utilize Multiple Small LoRAs to Outperform a Single Large LoRA?

**Motivation**

Our preceding analyses have characterized the fundamental properties of a single LoRA module. We have established that while its memory capacity is scalable with rank, it has discernible limits. Furthermore, we found that its parameter efficiency is often optimal in the low-to-mid rank regime. These findings motivate an alternative approach for managing large-scale knowledge, one that circumvents the limitations of a single, monolithic module. Specifically, the finite capacity of a single LoRA and the high parameter efficiency of low-rank modules suggest the potential advantages of a modular architecture: distributing knowledge across multiple small, efficient modules.

Therefore, this section conducts a proof-of-concept (PoC) to validate the core potential of this modular approach. To isolate the storage and retrieval benefits of the architecture itself, we perform this test within the PhoneBook (PB) dataset environment, where data length can be arbitrarily extended to simulate long-context challenges. Crucially, our initial analysis is performed under the assumption of an idealized setting with perfect routing. This allows us to evaluate the capacity of the modular system itself, free from any potential performance degradation that could be introduced by a separate routing mechanism.

**Experimental Setup**

This experiment was conducted on a 64K token PhoneBook dataset to compare three configurations under an identical parameter budget. The In-Context Learning (ICL) baseline was provided with the full 64K context at inference time. The single LoRA configuration used a single module with a rank of 32, trained on the entire 64K dataset. For the multi-LoRA configuration, the dataset was partitioned into eight 8K chunks, and a separate rank-4 module was trained on each chunk. For other experimental setups, we followed Appendix D.

**Experimental Results**

The results of this prototype experiment (Figure 6, left) highlight the potential of the modular approach for handling large volumes of information. On this long-context task, the ICL method exhibited a significant performance drop, while the single large LoRA approach struggled to internalize the extensive knowledge base effectively.

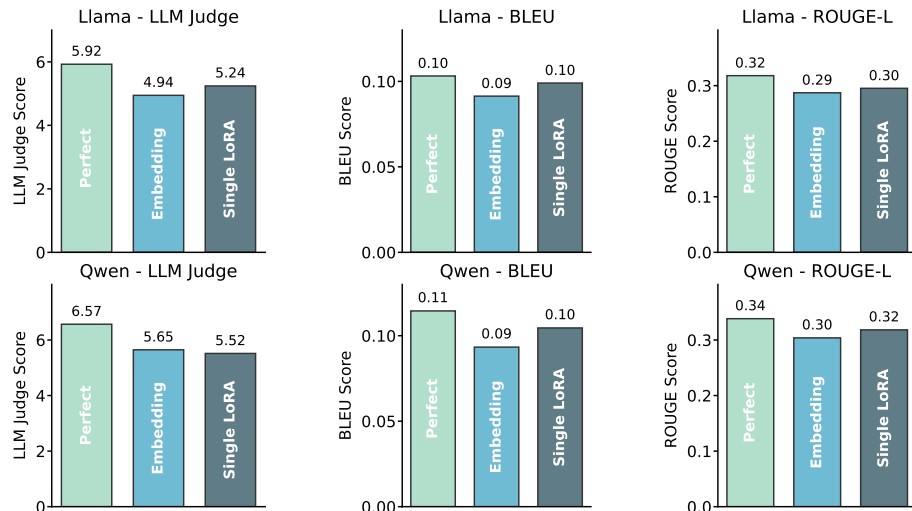

*Figure 19.* Performance comparison of routing strategies. **Top** Llama model results across LLM judge, BLEU, and ROUGE metrics. **Bottom** Qwen model results across LLM judge, BLEU, and ROUGE metrics.

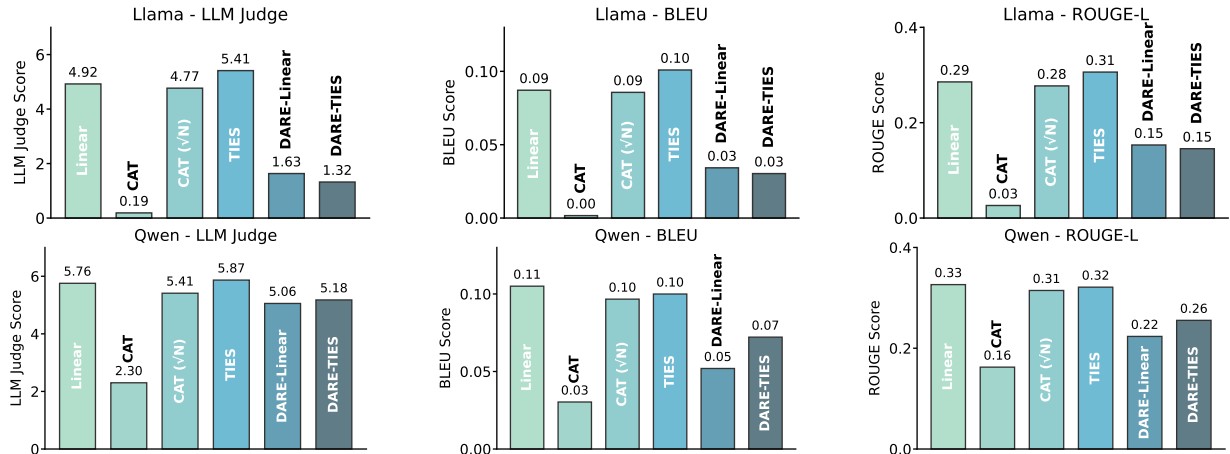

*Figure 20.* Performance comparison of different merging strategies. **(Top)** Results for Llama across LLM judge, BLEU, and ROUGE metrics. **(Bottom)** Results for Qwen across LLM judge, BLEU, and ROUGE metrics.

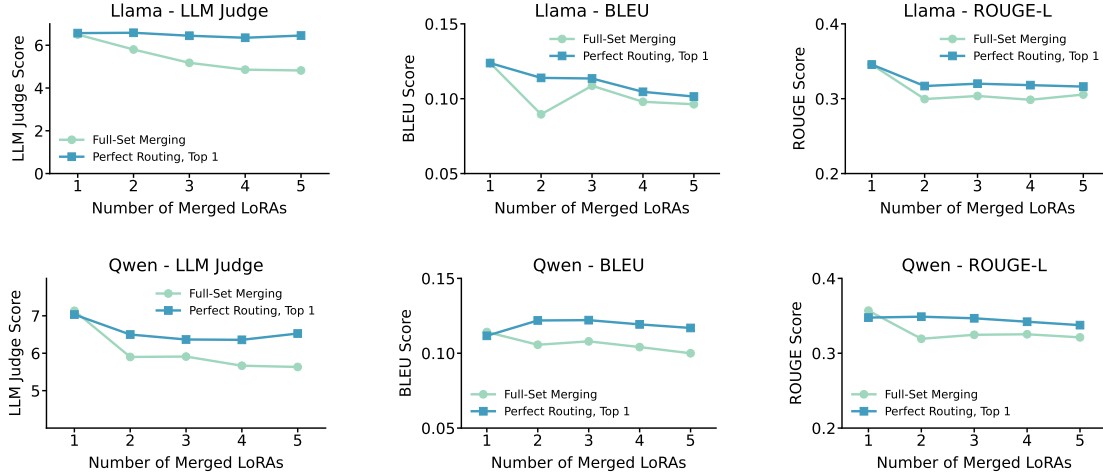

*Figure 21.* Comparison of Top-N performance. **(Top)** Llama results across LLM judge, BLEU, and ROUGE metrics. **(Bottom)** Qwen results across LLM judge, BLEU, and ROUGE metrics.

In contrast, the multi-LoRA system, operating under the perfect routing assumption, successfully learned the information within each respective chunk and demonstrated strong performance. While this outcome might be anticipated given the experimental design, its value lies not in its novelty but in its function as a clear proof of concept. This experiment serves to illustrate the conceptual advantages of a multi-LoRA architecture, where knowledge is partitioned and managed by specialized, efficient modules. Furthermore, it provides a foundational basis for discussing the critical components, such as the routing mechanism, that must be addressed to translate this concept into a practical, real-world system.

## N. Details of Q9. How Much Performance Loss Does a More Practical Routing Incur?

**Motivation**

The preceding proof-of-concept experiment demonstrated the potential of a multi-LoRA system under the idealized condition of perfect routing. However, real-world systems must make decisions based on imperfect information. This section aims to quantify the performance gap between this ideal scenario and a more practical implementation using embedding-based retrieval. Measuring this gap is essential for understanding the performance cost imposed by real-world constraints.

Furthermore, this analysis helps to identify critical performance bottlenecks in a modular system. The final performance of a multi-LoRA system depends on multiple components, and this experiment investigates the extent to which the routing stage can limit the system's overall efficacy. If the performance loss from standard retrieval-based routing proves to be substantial, it would highlight the need for alternative strategies—such as merging multiple LoRAs—to mitigate the inherent uncertainties of the routing process.

**Experimental Setup**

To evaluate the impact of the routing mechanism, we compared the following three approaches.

**Perfect Routing (Oracle).** This serves as the theoretical upper bound for system performance. For each query, an oracle, with prior knowledge of which document chunk (e.g., a specific paper in the PaperQA dataset) contains the answer, selects the correct corresponding LoRA module.

**Embedding-based Routing.** For the practical text-embedding-based routing scenario, we employed the `BGE-large-en-v1.5` (Chen et al., 2024) model to select LoRA modules based on embedding similarity between the query and source documents; this model is used for all subsequent embedding-based operations unless otherwise specified.

**Single LoRA (Baseline).** For comparison, we also include the performance of a single, monolithic LoRA module trained on the entire dataset.

For other experimental setups, we followed Appendix D.

**Experimental Results**

Figure 6 (right) and Figure 19 show that the choice of routing mechanism is a critical factor in the performance of a multi-LoRA system. As expected, the perfect routing approach yielded the highest performance, which reaffirms that the modular approach can be highly effective if the correct specialized module is accurately identified. In contrast, the embedding-based routing method showed a noticeable performance degradation compared to the perfect routing oracle. This performance gap suggests that the retrieval-based routing mechanism is a primary limiting factor for the system's overall potential.

Perhaps the most notable finding is that the Embedding-based Routing approach performed at a lower level than the single LoRA baseline. This outcome is particularly insightful, as it indicates that selecting an incorrect, highly specialized LoRA module can be more detrimental to performance than using a single, non-specialized module that contains the entirety of the knowledge base. This underscores the critical importance of routing accuracy in a modular architecture.

**Additional Routing Experiments: Token-Based Routing**

The main Q9 experiment uses embedding-based routing with `BGE-large-en-v1.5`, which is a standard retrieval-style choice for selecting relevant LoRA modules. To test whether the observed routing bottleneck is specific to embedding-

*Table 6.* Additional routing experiments comparing embedding-based routing with token-based LoRA routers. NarrativeQA is evaluated with ROUGE-L using Qwen3-8B over 40 documents. QuALITY is evaluated with accuracy using Llama-3.1-8B over 115 articles. Top-3 results use TIES merging after routing.

| Routing | NarrativeQA | | QuALITY | |
|---|---|---|---|---|
| | Top-1 | Top-3 | Top-1 | Top-3 |
| Arrow | **21.93** | 21.15 | 33.68 | 39.38 |
| SpectR | 19.42 | **21.33** | 35.59 | 42.13 |
| LAG | 18.34 | 21.25 | 36.28 | 41.71 |
| Embedding | 19.13 | 21.15 | **37.84** | **42.42** |

based retrieval, we additionally evaluate three token-based LoRA routing methods: Arrow (Ostapenko et al., 2024b), SpectR (Fleshman & Van Durme, 2025b), and LAG (Fleshman & Van Durme, 2025a). We compare these methods against the embedding-based router on two long-document settings: NarrativeQA with Qwen3-8B, evaluated by ROUGE-L, and QuALITY with Llama-3.1-8B, evaluated by accuracy. For each router, we report both Top-1 routing and Top-3 routing followed by TIES merging.

Table 6 shows that token-based and embedding-based routing perform comparably in most settings. Arrow improves Top-1 routing on NarrativeQA, suggesting that token-level matching can help under strict single-module selection in free-form QA. However, no token-based method consistently dominates the embedding-based baseline across datasets and retrieval depths. This suggests that routing remains a system-level bottleneck for multi-LoRA memory, and that improving robustness likely requires considering routing together with partitioning granularity and merging interference.

## O. Details of Q10. Can Merging Multiple LoRAs Mitigate Routing Error?

**Motivation**

The previous section established that retrieval-based, top-1 routing can be a significant performance bottleneck, highlighting the inherent risks of relying on a single module selection. This finding motivates the exploration of an alternative strategy to distribute this risk: retrieving the top-k relevant LoRA modules and merging their knowledge. This section aims to validate the feasibility of such a merging strategy as a direct approach to mitigating routing uncertainty.

Furthermore, LoRA merging is not a monolithic concept; it encompasses a design space ranging from simple arithmetic combinations to more sophisticated algorithms intended to mitigate interference between parameters. We investigate how different merging algorithms perform from a knowledge preservation perspective and analyze the reasons for these differences. This inquiry also touches upon a fundamental property of LoRAs: their composability. By arithmetically combining independently trained modules, we can assess whether the knowledge encoded in each can be preserved without destructive interference, thereby testing their viability as composable building blocks.

**Experimental Setup**

We provide a detailed description of the five LoRA merging methods evaluated in this section: (1) Linear Averaging, (2) Matrix Concatenation (CAT), (3) Scaled Concatenation (CAT-$1/\sqrt{N}$), (4) TIES-Merging, and (5) DARE.

**Linear Averaging.** Linear Averaging is the most intuitive and straightforward method for merging multiple LoRA modules. The merge operates directly on the low-rank factors $A_i$ and $B_i$ rather than on the materialized delta $\Delta W_i = B_i A_i$. With uniform weighting over $N$ LoRA modules, each factor is rescaled by $1/N$ before summation:

$$A_{\text{merged}} = \frac{1}{N} \sum_{i=1}^{N} A_i, \qquad B_{\text{merged}} = \frac{1}{N} \sum_{i=1}^{N} B_i.$$

While this method is simple to implement, its primary limitation is its inability to account for potential conflicts or redundancies among the parameters of the different LoRA modules.

**Matrix Concatenation (CAT).** CAT constructs a single, more expressive LoRA module by concatenating the respective low-rank matrices ($A$ and $B$) from multiple source modules, effectively increasing the rank of the final merged adapter. Given $N$ LoRA modules of rank $r$ with $A_i \in \mathbb{R}^{d \times r}$ and $B_i \in \mathbb{R}^{r \times k}$, the CAT merge combines them along the rank

dimension:
$$A_{\text{merged}} = \text{concat}([A_1, \ldots, A_N], \dim = 1), \quad B_{\text{merged}} = \text{concat}([B_1, \ldots, B_N], \dim = 0).$$

The resulting merged adapter has rank $Nr$, with full-weight delta

$$\Delta W_{\text{CAT}} = B_{\text{merged}} A_{\text{merged}} = \sum_{i=1}^{N} B_i A_i,$$

which is the sum—rather than the average—of the individual deltas.

**Scaled Concatenation (CAT-$1/\sqrt{N}$).** A key issue with vanilla CAT is that it sums the individual deltas rather than averaging them, injecting an $N\times$ over-perturbation to the base model. Scaled CAT corrects this by rescaling each LoRA factor by $1/\sqrt{N}$ before concatenation:

$$\tilde{A}_i = \tfrac{1}{\sqrt{N}} A_i, \qquad \tilde{B}_i = \tfrac{1}{\sqrt{N}} B_i.$$

Concatenating these rescaled factors yields a merged adapter of rank $Nr$ whose full-weight delta is the exact mean of the individual deltas:

$$\Delta W_{\text{CAT-}1/\sqrt{N}} = \sum_{i=1}^{N} \tilde{B}_i \tilde{A}_i = \frac{1}{N} \sum_{i=1}^{N} B_i A_i.$$

**TIES.** TIES (TrIm, Elect sign, and Merge) (Yadav et al., 2023) is an advanced methodology designed to mitigate the interference that often occurs when merging parameters from multiple fine-tuned models. It specifically addresses two primary sources of interference: parameter redundancy and sign conflicts. The TIES-Merging procedure consists of the following three steps:

1. **Trim:** This step zeroes out parameters that underwent minimal change during the fine-tuning of each LoRA module. By retaining only the parameters with the largest change in magnitude (e.g., the top-k percentile), this process filters out redundant or less impactful parameters.

2. **Elect Sign:** This step resolves directional conflicts in parameter updates across different LoRA modules. For a given parameter, some LoRAs may have induced a positive update while others induced a negative one. TIES elects a single, dominant sign based on a majority vote, where the sign corresponding to the greatest total magnitude of updates is chosen as the consensus direction.

3. **Merge:** Finally, only the parameter values that align with the elected sign are averaged. Parameters with conflicting signs are discarded from the merge for that specific weight, thus minimizing negative interference.

**DARE (Drop And REscale).** To address the extreme redundancy in delta parameters, we employ DARE (Yu et al., 2024), which randomly drops a fraction $p$ of the parameters and rescales the remaining ones to approximate the original embeddings. Formally, for a given LoRA weight matrix $W$, the sparsified weight $\hat{W}$ is computed as:

$$\hat{W} = \frac{1}{1-p}(W \odot M)$$

where $M \in \{0, 1\}^d$ is a binary mask sampled from a Bernoulli distribution with parameter $1 - p$. We integrate this sparsification as a pre-processing step to mitigate parameter interference, resulting in two variants: **DARE-Linear** and **DARE-TIES**. DARE-Linear performs standard linear averaging on the sparsified modules, while DARE-TIES utilizes DARE-processed weights as input for the sign election and merging stages of TIES.

For other experimental setups, we followed Appendix D.

**Experimental Results**

Our results indicate that the choice of merging algorithm critically affects the performance of the multi-LoRA system. According to Figure 7 (left) and Figure 20,

both the **TIES** and **Linear** merging methods demonstrated the most robust performance across all architectures. Notably, the naive Linear merge proved to be a surprisingly strong baseline, achieving results comparable to TIES, particularly in

the Qwen model. However, TIES consistently maintained a slight performance edge and greater stability across different settings. Its score surpassed that of the top-1 Embedding-based routing approach and was nearly on par with the single LoRA baseline. This suggests that while simple averaging is often sufficient for retaining memorized information, the interference resolution mechanisms in TIES offer a marginal but valuable advantage in stabilizing multi-LoRA composition.

In contrast, **DARE**-based approaches failed to match the performance of these baselines. This shortfall is likely due to DARE's random parameter dropping strategy. In a task that requires the precise memorization of dense information, randomly pruning weights carries a high risk of discarding critical knowledge fragments. Unlike TIES, which preserves the most significant parameters based on magnitude, DARE's stochastic nature appears ill-suited for maintaining the fidelity of specific memory traces.

The **CAT** method exhibited the lowest performance by a wide margin, but this collapse turns out to stem from an avoidable cause rather than a structural one. By summing the individual deltas rather than averaging them, vanilla CAT applies an $N\times$ over-perturbation that distorts the base model's output distribution. The **CAT-**$1/\sqrt{N}$ variant rescales each LoRA factor by $1/\sqrt{N}$ so that the merged delta becomes the mean of the individual deltas while preserving the expanded rank $Nr$, and empirically performs comparably to Linear. This confirms that scale mismatch—not rank expansion—is CAT's dominant failure mode. The residual gap from CAT-$1/\sqrt{N}$ to TIES further indicates that, once magnitude is calibrated, the remaining cost reflects unresolved parameter interference: conflicting updates that cancel destructively under arithmetic merging but are explicitly reconciled by TIES through trimming and sign election. Taken together, these findings refine the common view that concatenation is intrinsically unsuitable for LoRA composition; once calibrated, it is competitive with linear averaging, and the gap to TIES isolates the cost of unmanaged interference.

## P. Details of Q11. How Does the Number of Merged LoRAs Affect Performance?

**Motivation**

In a system that merges the top-N retrieved LoRAs, the choice of N is a critical hyperparameter that mediates a trade-off between the *recall* of the routing phase and the *precision* of the merging phase. Understanding how system behavior changes with N is essential for designing and tuning practical multi-LoRA architectures.

This system architecture inherently faces two competing risks: (1) **routing failure**, where a small N may fail to retrieve the correct knowledge module, and (2) **merging failure**, where a large N may lead to knowledge corruption through parameter interference. Our previous experiments provided evidence for both: the superior performance of a top-3 TIES merge over a top-1 selection suggested that merging can mitigate routing failures, while the poor performance of linear merging hinted at the dangers of parameter interference. This section seeks to empirically map this trade-off by systematically varying N, providing an analysis of how sensitively parameter interference affects the system as the number of merged modules grows.

**Experimental Setup**

The experiment was designed to observe performance changes as we increased the number of merged LoRA modules, N, from 1 to 5. To isolate the effect of the merging process from routing errors, our evaluation for a given N involved merging the specific N LoRA modules that were trained on the N documents from which the test queries were sampled. This setup ensures that all necessary knowledge is present within the merged set, allowing us to measure only the degradation caused by the merging process itself. For other experimental setups, we followed Appendix D.

**Experimental Results**

The results show a consistent and sharp decline in overall system performance as the number of merged LoRAs (N) increases. As illustrated in Figure 7 (right) and Figure 21, the performance curve peaks at N=1 (which is equivalent to perfect top-1 routing) and then degrades steadily as N grows.

This outcome provides strong empirical evidence for the phenomena of knowledge dilution and parameter interference. The analysis suggests that as more LoRA modules—even relevant ones—are merged, the cumulative conflicts between their parameter values rapidly degrade system performance.

These findings reveal a distinct trade-off in multi-LoRA systems that is governed by the number of merged modules. A small N risks routing failure, whereas a large N is dominated by performance loss from parameter interference. This implies that a

*Table 7.* Additional results on 100K+ token benchmarks. ∞Bench is evaluated with ROUGE-L over 20 documents longer than 100K tokens. LongBench v2 is evaluated with accuracy over 30 documents of length 101K–364K tokens.

| Method | ∞Bench | | LongBench v2 | |
|---|---|---|---|---|
| | Qwen3-8B | Qwen3-1.7B | Qwen3-8B | Qwen3-1.7B |
| *Closed Book* | | | | |
| Single LoRA | **24.37** | 17.91 | **30.00** | 23.33 |
| Multi-LoRA Top3 | 23.49 | **20.41** | **30.00** | **36.67** |
| *Open Book* | | | | |
| ICL | 30.89 | 14.30 | 20.00 | 20.00 |
| RAG | 32.80 | 30.46 | 20.00 | 16.67 |
| Multi-LoRA Top3 + ICL | **34.92** | 29.60 | **33.33** | 26.67 |
| Multi-LoRA Top3 + RAG | 32.10 | **31.39** | 16.67 | **30.00** |

successful multi-LoRA system requires a more sophisticated strategy than simply merging a fixed number of top-retrieved modules. Future work should explore methods for dynamically adjusting N based on retrieval confidence or developing merging algorithms that are more robust to interference, thereby striking a more effective balance between routing and merging.

## Q. Long-Document Benchmarks

This appendix details the long-document case study settings used in Section 7 (Q12–Q15), including the NarrativeQA and QuALITY benchmarks.

**NarrativeQA (NQA).** NarrativeQA (Kočiskỳ et al., 2018) is a long-context question answering benchmark with very long narratives (often far beyond typical context windows). The official dataset is split into 1,102 training documents, 115 development documents, and 355 test documents. Questions are written based on human-written summaries of the entire text, while evaluation targets the original full narrative; consequently, answers may require synthesizing information scattered across the document.

To keep the case study computationally focused while preserving long-context characteristics, we restrict to documents whose tokenized length is between 10K and 20K tokens. We randomly sample 40 documents (with their associated QA pairs) from the official training and development splits.

**QuALITY.** QuALITY (Pang et al., 2022) consists of long-form articles paired with multiple-choice questions. We use the validation split (115 articles) without additional length filtering. We report accuracy on the multiple-choice questions. We evaluate NarrativeQA with multi-reference ROUGE-L. We evaluate QuALITY with multiple-choice accuracy on the validation split.

### Additional 100K+ Token Benchmarks

To address whether LoRA-based memory remains viable beyond the 10K–20K token regime considered in NarrativeQA and QuALITY, we additionally evaluate on two extreme long-context benchmarks: ∞Bench (Zhang et al., 2024a) and LongBench v2 (Bai et al., 2025). For ∞Bench, we use 20 documents exceeding 100K tokens and report ROUGE-L. For LongBench v2, we use 30 documents ranging from 101K to 364K tokens and report accuracy. Table 7 shows that LoRA-based methods do not uniformly collapse in the 100K+ token regime; instead, their effectiveness remains setting-dependent, with hybrid variants often recovering or improving performance. This supports our conclusion that LoRA is best viewed as a complementary memory component rather than a standalone replacement for ICL or RAG.

## R. Details of Q12. How Does LoRA Memory Perform on Long-Context Multi-Hop QA Task?

### Motivation

While our preceding experiments utilized datasets with explicitly partitioned corpora (e.g., PhoneBook, PaperQA), real-world scenarios often involve long-form documents without clear boundaries. To investigate this, we use the NarrativeQA (NQA) dataset (Kočiskỳ et al., 2018), which consists of long narratives where segmenting the text for individual LoRAs creates

strong contextual dependencies between chunks. The NQA questions often require synthesizing information across these chunk boundaries, demanding a holistic understanding of the text. This inter-chunk dependency presents a critical challenge for modular approaches, as a multi-LoRA system may fail to capture the overarching narrative. This section analyzes this potential failure mode and its impact on performance.

### Experimental Setup

To assess the standalone capability of LoRA as a memory for long-context and multi-hop reasoning, we evaluated different parametric memory configurations using NarrativeQA and QuALITY benchmarks across four base models (Llama-3.2-1B, Llama-3.1-8B, Qwen3-1.7B, and Qwen3-8B). We compared two primary architectural approaches. First, the single LoRA setup trains a monolithic module ($r = 16$) on the entire document text. Second, the multi-LoRA setup partitions the document into chunks (2,048 tokens for NQA, 768 tokens for QuALITY) and trains a separate, smaller module ($r = 4$) for each chunk. For multi-LoRA inference, we retrieve the most relevant modules using embedding similarity and merge them via TIES-Merging, evaluating both Top-1 and Top-3 settings.

**Baselines.** We benchmarked these LoRA systems against the raw Base Model (zero-shot), standalone ICL, standalone RAG and SDCD (Caccia et al., 2025).

For other experimental setups, we followed Appendix D.

### Experimental Results

Although a multi-LoRA system is conceptually well-suited for such long documents, the observed performance deficit is likely a consequence of the compounding error sources discussed in Sections Q9 and Q10. The dual challenges of routing inaccuracy and parameter interference are intensified by the nature of the NarrativeQA dataset. Its reliance on multi-hop questions, which demand the synthesis of information across multiple text segments, inherently strains the capabilities of our chunk-based modular approach.

## S. Details of Q13. Can LoRA Perform Better When Used With External Context?

### Motivation

LoRA-based models store knowledge in a compressed format, which means they do not have access to the original, high-fidelity training data at inference time. This inherent characteristic can be a limitation. The challenge is further compounded in multi-LoRA systems. This problem is exacerbated in multi-LoRA systems, where training across multiple modules can lead to a loss of contextual continuity and fragmented internal memory.

This raises a critical question: *can supplying explicit external context at inference time compensate for these limitations?* In this section, we investigate the potential of augmenting both single and multi-LoRA systems with external information. We hypothesize that such an approach will enhance overall performance by providing the models with precise, relevant information that may be absent or diluted in their internal representations. We further posit that this benefit will be particularly pronounced for multi-LoRA configurations, as external context can help bridge the narrative gaps introduced by the module merging process.

### Experimental Setup

To evaluate the interaction between LoRA and external context, we constructed several hybrid systems and benchmarked them against standalone baselines.

**Hybrid Systems.** We explored two primary integration strategies. First, we paired a single LoRA module with both In-Context Learning (ICL) and Retrieval-Augmented Generation (RAG). Second, given its architectural alignment with retrieval mechanisms, we combined our chunk-based multi-LoRA system with RAG. For the multi-LoRA hybrid, we evaluated two distinct merging strategies: one that merges the single most relevant LoRA module retrieved (Top-1) and another that merges the top three most relevant modules (Top-3).

**Baselines.** The performance of these hybrid systems was compared against standalone ICL, standalone RAG, and the closed-book single and multi-LoRA configurations from our previous experiments.

For other experimental setups, we followed Appendix D.

**Experimental Results**

As presented in Table 2, the integration of LoRA with external context improves performance across all configurations. For both model scales, the hybrid systems outperform their standalone LoRA, ICL, and RAG counterparts.The performance gain is most pronounced for the multi-LoRA system. This suggests that the provision of external information effectively compensates for the precision loss that can occur when merging multiple specialized LoRA modules.

Interestingly, we observe that ICL yields a greater performance uplift than RAG when combined with LoRA. A possible explanation is that the continuous, global context supplied by ICL is uniquely effective at restoring the narrative cohesion that may be lost among fragmented LoRA modules. The snippet-based retrieval approach of RAG, while beneficial, appears less capable of fully overcoming this particular challenge.

## T. Details of Q14. Can Merging LoRAs Improve Contextual Continuity?

**Motivation**

In the previous section, we demonstrated that external context can help mitigate the loss of contextual continuity in multi-LoRA systems. An alternative approach for complex reasoning involves the direct synthesis of knowledge from multiple LoRA modules. PRAG (Su et al., 2025) has shown that merging LoRAs can be effective for tasks requiring the integration of knowledge from several distinct sources. However, PRAG focused on scenarios where the knowledge is sourced from multiple, explicitly distinct, and relatively short documents. The question of whether such a merging strategy is beneficial for tasks demanding deep, continuous comprehension of a single long document has not been investigated. This motivates our current investigation: we re-evaluate the efficacy of the multi-LoRA merging strategy on the NarrativeQA dataset, which requires holistic document comprehension. This allows for an assessment of whether synthesizing knowledge from multiple modules offers a tangible advantage over relying on a single, best-matched module for multi-hop QA tasks.

**Experimental Setup**

The overall experimental setup is identical to Q12 and Q13.

**Experimental Results**

The results, presented in Table 2, reveal a clear and consistent performance advantage for the Top-3 merging strategy over the Top-1 approach. In the closed-book setting, the larger model in particular shows an improvement with the Top-3 configuration compared to the Top-1.

This performance gap becomes more pronounced in the open-book setting, where models are provided with external context. The multi-LoRA Top-3 configurations, when paired with either ICL or RAG, consistently and significantly outperform their Top-1 counterparts. Notably, the combination of Top-3 merging with ICL achieves the highest performance across all tested methods.

This finding suggests that for tasks requiring multi-hop reasoning, accessing a broader set of knowledge sources by merging multiple modules is highly advantageous. The model appears to leverage the combined knowledge from several specialized LoRAs to construct more comprehensive and accurate answers, a benefit that is amplified by the presence of external guiding context.

## U. Details of Q15. How does LoRA benefit in time?

**Motivation**

While the main paper establishes the performance of LoRA-based memory systems in terms of accuracy, a comprehensive evaluation must also consider their practical viability from a computational standpoint. Context-based methods like ICL and RAG are known to incur significant latency due to the need to process long context windows for every query. In contrast, LoRA-based methods internalize knowledge into parameters, theoretically enabling much faster inference with short contexts, but introducing their own overheads such as module loading and merging.

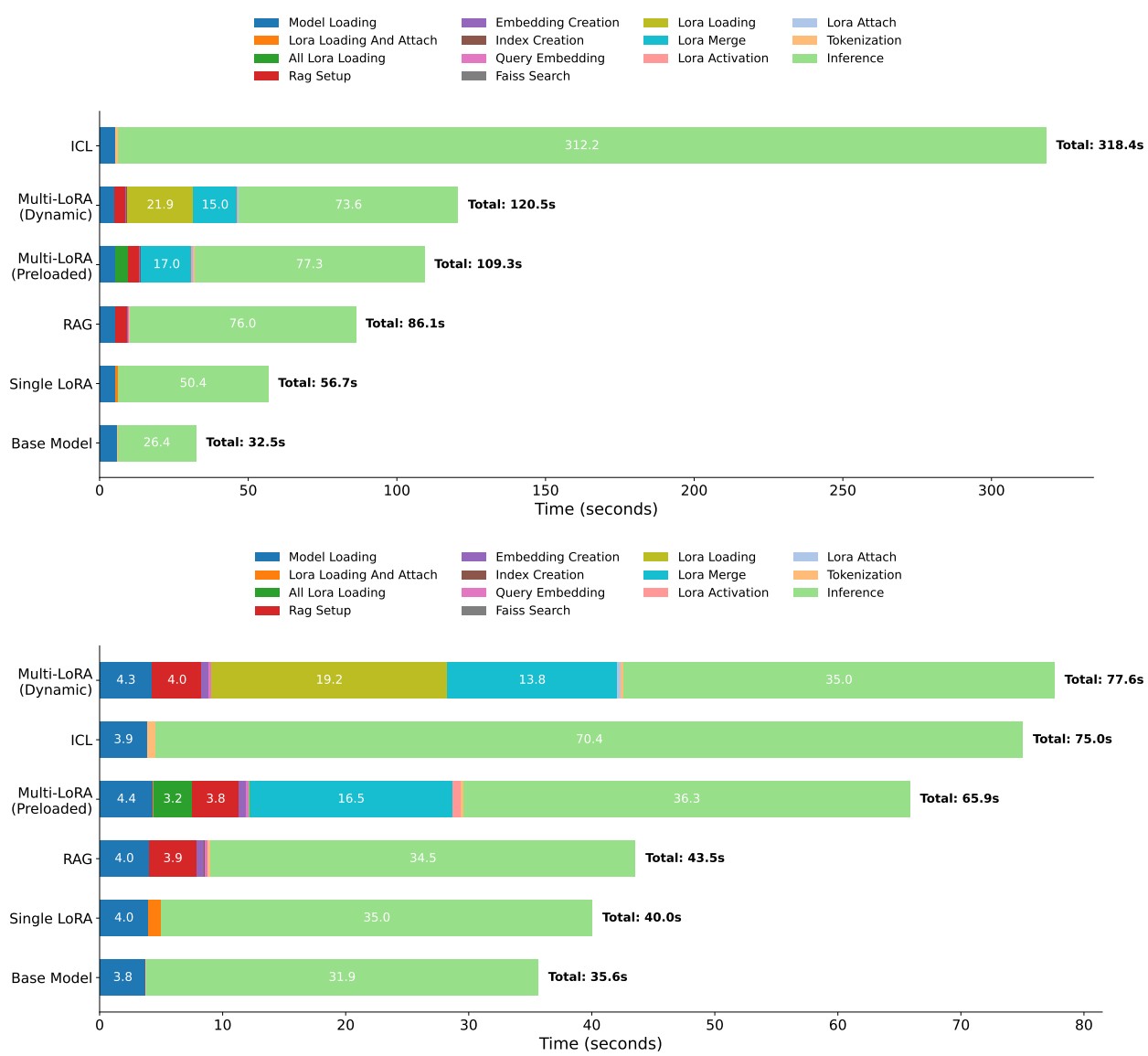

*Figure 22.* Comparison of method execution times. (**Top**) Without Flash Attention. (**Bottom**) With Flash Attention.

The purpose of this experiment is to provide a granular, quantitative analysis of these computational trade-offs. By measuring the end-to-end latency and breaking down the time spent on each component—from model loading to final token generation—we can create a clear picture of the real-world costs and benefits associated with each approach. This section details the complete methodology for this analysis, beginning with the hardware and software environment, followed by the rigorous timing measurement protocol, and concluding with a precise definition of every component measured for each experimental condition.

### Experimental Setup

**Hardware and Software Specifications.** All experiments were conducted on RTX PRO 6000 Blackwell. The core software stack included PyTorch for model operations, the Hugging Face `transformers` library for loading the base model, the PEFT (Mangrulkar et al., 2022) library for handling LoRA modules, `sentence-transformers` for text embeddings, and `faiss` for efficient similarity search in retrieval tasks. The base model for all experiments was `meta-llama/Llama-3.1-8B-Instruct`, and the retrieval model was `BAAI/bge-large-en-v1.5`.

**Timing Measurement Protocol.** To ensure accurate and reliable timing, especially for GPU operations, we implemented a standardized measurement protocol. All timed operations were wrapped in a utility function that uses `time.perf_counter()` for high-precision timing. Crucially, before starting and after ending the timer, we explicitly called `torch.cuda.synchronize()`. This call blocks the CPU execution until all previously queued GPU kernels have completed, which is essential for accurately measuring the wall-clock time of asynchronous GPU computations and avoiding misleading results.

**Methodology and Measured Components**

We systematically measured the time taken by each distinct stage of the inference process for six different methods. The total time for each method is the sum of its one-time setup costs and the cumulative time of all per-question operations over each NQA document's 30 questions in the evaluation set.

BASE MODEL (CLOSED-BOOK)

This configuration measures the baseline performance of the LLM without any external context or adapters.

- **model_loading**: One-time cost to load the weights of the `Llama-3.1-8B-Instruct` model into GPU memory.

- **tokenization**: Per-question time to convert the short prompt (question only) into input tokens.

- **inference**: Per-question time for the model to generate the answer tokens.

IN-CONTEXT LEARNING (ICL)

This method provides the model with the entire source document as context for every query.

- **model_loading**: Same as the base model.

- **tokenization**: Per-question time to tokenize the prompt, which includes the **full document text** and the question. This is computationally more expensive than in the closed-book setting.

- **inference**: Per-question generation time. This is the most significant component due to the quadratic complexity of self-attention over the long context window.

RETRIEVAL-AUGMENTED GENERATION (RAG)

This method retrieves relevant text snippets to use as context.

- **model_loading**: One-time cost to load the LLM.

- **rag_setup**: A one-time setup cost that includes:
  - Loading the `bge-large-en-v1.5` embedding model.
  - **embedding_creation**: Generating embeddings for all text chunks of the document.
  - **index_creation**: Building the FAISS index from the chunk embeddings.

- **query_embedding**: Per-question time to generate an embedding for the input question.

- **faiss_search**: Per-question time to perform a similarity search against the FAISS index to retrieve the top-3 relevant chunks.

- **tokenization**: Per-question time to tokenize the prompt containing the retrieved chunks and the question.

- **inference**: Per-question generation time using the retrieved context.

SINGLE LORA (CLOSED-BOOK)

This method uses a single LoRA module trained on the entire document.

- **model_loading**: One-time cost to load the base LLM.

- **lora_loading_and_attach**: A one-time setup cost to load the single LoRA adapter from disk and attach it to the base model using `PeftModel.from_pretrained`.

- **tokenization**: Per-question time to tokenize the short prompt (question only).

- **inference**: Per-question generation time.

MULTI-LORA (PRELOADED)

This method preloads all LoRA modules for a document into GPU memory at the start and merges the relevant ones for each query. This strategy is designed to minimize I/O latency during inference.

- **model_loading**: One-time cost to load the base LLM.

- **all_lora_loading**: A one-time setup cost to load **all** LoRA modules corresponding to the document's chunks into GPU memory.

- **rag_setup**: Same as the RAG method, used for retrieving relevant LoRA modules.

- **query_embedding** and **faiss_search**: Per-question retrieval time to identify the top-3 relevant LoRA modules.

- **lora_merge**: Per-question time to combine the weights of the top-3 retrieved LoRA adapters into a new, temporary adapter using the TIES-merging algorithm (`model.add_weighted_adapter`).

- **lora_activation**: Per-question time to set the newly merged adapter as the active one for inference (`model.set_adapter`).

- **tokenization** and **inference**: Per-question time for generation using a short prompt.

MULTI-LORA (DYNAMIC)

This method loads the required LoRA modules from disk for each query, representing a scenario where preloading is not feasible.

- **model_loading**: One-time cost to load the base LLM.

- **rag_setup**: Same as the RAG method.

- **query_embedding** and **faiss_search**: Per-question retrieval time to identify relevant LoRAs.

- **lora_loading**: Per-question time to dynamically load the top-3 retrieved LoRA adapters from disk into memory. This represents a significant I/O overhead.

- **lora_merge** and **lora_attach**: Per-question time to merge the newly loaded adapters and set the result as active.

- **tokenization** and **inference**: Per-question generation time.

For other experimental setups, we followed Appendix D.

### Experimental Results

The results of the timing experiment, as visually detailed in the stacked bar chart in Figure 22, confirm several initial hypotheses while also revealing unexpected interactions between LoRA modules and underlying model optimizations. The figure provides a clear breakdown of total execution time into its constituent components for each method, which are analyzed below.

**Overall Performance Comparison.** As hypothesized, the In-Context Learning (ICL) method was by far the most time-consuming. Its total processing time was dominated by the inference stage, a direct consequence of the substantial computational cost of applying self-attention over the full document context for every query. Among the LoRA-based methods, a key finding is the significant advantage of the preloading strategy. The multi-LoRA (Preloaded) configuration, which loads all necessary adapters into GPU memory once at startup, was considerably faster than the multi-LoRA (Dynamic) approach. The latter was bottlenecked by the per-query I/O latency of loading adapters from disk. This result underscores that while LoRA offers efficient inference, the surrounding architecture for managing and serving modules has significant potential for optimization.

**Inference Time and Flash Attention Interactions.** A deeper analysis of the inference component, particularly concerning the use of Flash Attention, yielded more intriguing results. When experiments were run with Flash Attention disabled, the performance aligned with general expectations for context-heavy methods; ICL's inference time was overwhelmingly longer than all other methods. However, an unexpected discrepancy emerged between the closed-book methods. The inference time for the single LoRA configuration was approximately twice as long as that of the base model, despite both processing identical short-context prompts (i.e., the question only). Theoretically, the raw computational cost for inference should be nearly identical, as the number of floating-point operations is not significantly increased by the LoRA adapter. This discrepancy strongly suggests that the presence of the PEFT adapter may inadvertently prevent the underlying model from activating certain default optimization strategies during its forward pass. Conversely, when experiments were run with Flash Attention enabled, another counterintuitive trend was observed. The inference speed of the base model paradoxically decreased compared to its non-Flash-Attention run, while the inference speed for the single LoRA model significantly improved, becoming faster than the base model.

