# OpenReview forum: "Understanding LoRA as Knowledge Memory: An Empirical Analysis"
_ICML.cc/2026/Conference — ICML 2026 regular_

### Official Review · Reviewer_aR9x · 2026-02-24

**Soundness:** 3
**Presentation:** 4
**Significance:** 3
**Originality:** 2
**Overall Recommendation:** 5
**Confidence:** 5

**Summary:**

This paper provides an empirical study of the characteristics and practicality of using LoRA to augment LMs with knowledge. Several questions are raised and studied, providing insight into LoRA capacity constraints, efficiency, single vs. multi-LoRA setups, and the interactions between parameteric and in-context knowledge. The experiments use a wide range of models, datasets (including the introduction of a PaperQA benchmark), and LoRA methods, providing important considerations for practitioners using LoRA to augment a model with additional knowledge.

**Compliance With Llm Reviewing Policy:**

Affirmed.

**Final Justification:**

I recommend accepting this paper. The discussion period reinforced my positive score.

**Key Questions For Authors:**

1) Were hyperparameters such as LoRA alpha and learning rate tuned per experiment? For example, when sweeping ranks from 2-1024, were the hyperparameters tuned for each?

**Limitations:**

yes

**Strengths And Weaknesses:**

**Strengths**

Soundness:

1) The paper does an excellent job targeting specific research questions with careful experimentation, ablation, and interpretation of results.

Presentation:

2) The paper is well structured, with clear and concise segments per research question.
3) The figures are easily interpretable and visually appealing.
4) The implications are highlighted for casual readers, with a good balance of detail in the main body and additional specifics in the appendix.

Significance:

5) The paper addresses important questions driving the active research areas surrounding knowledge-augmentation with LoRA and multi-LoRA system development and deployment.
6) The experiments provide practical takeaways, immediately useful to ongoing and future research directions.

Originality:

7) While the paper exclusively studies existing methods, it comprehensively covers important aspects and considerations commonly underexplored in the literature.

**Weaknesses**

Soundness:

1) A broader set of methods would more accurately answer some of the posed questions. For example, only embedding-based retrieval was used in multi-LoRA setups, and the retrieved adapters were applied globally. The paper cites at least two token-based routing approaches (Arrow and LAG), which would provide a different perspective for practical routing and merging results.

Presentation:

N/A

Significance:

2) Some of the questions and implications target well-known characteristics of LoRA, such that capacity increases with rank and will be finite, or that synthetic data improves knowledge acquisition. This is only a minor point, as including these provides comprehensive coverage.

Originality:

3) As an empirical study, the paper covers existing approaches. There are avenues for the authors to increase originality, for example, they could synthesize the implications into a new muli-LoRA system or study theoretical aspects of some of the findings: information-theoretic capacity constraints and scaling laws come to mind.

---

> ### Author Rebuttal · Authors · 2026-03-31
>
> We sincerely thank the reviewer for the positive and thorough evaluation, and especially the high-confidence assessment. We address each point below.
>
> ---
>
> ## W1. Token-based routing methods for multi-LoRA setups
>
> We thank the reviewer for this constructive suggestion. Following your recommendation, we conducted additional experiments with three token-based routing methods—Arrow (Ostapenko et al., 2024), SpectR (Fleshman & Van Durme, 2025a), and LAG (Fleshman & Van Durme, 2025b)—alongside our embedding-based baseline.
>
> **NarrativeQA (40 docs, Qwen3-8B, ROUGE-L):**
>
> | Routing | Top-1 | Top-3 |
> |---------|-------|-------|
> | Arrow | **21.93** (+2.80) | 21.15 |
> | SpectR | 19.42 (+0.30) | **21.33** (+0.20) |
> | LAG | 18.34 (−0.80) | 21.25 (+0.10) |
> | Embedding | 19.13 | 21.15 |
>
> **QuALITY (115 articles, Llama-3.1-8B, Accuracy):**
>
> | Routing | Top-1 | Top-3 |
> |---------|-------|-------|
> | Arrow | 33.68 (−4.16) | 39.38 (−3.04) |
> | SpectR | 35.59 (−2.25) | 42.13 (−0.29) |
> | LAG | 36.28 (−1.56) | 41.71 (−0.71) |
> | Embedding | **37.84** | **42.42** |
>
> Overall, token-based and embedding-based routing perform comparably in most settings, with differences largely within 1–2%p. The one notable exception is Arrow Top-1 on NarrativeQA (+2.8%p over embedding), suggesting a potential benefit of token-level matching under strict single-module selection in free-form QA. Embedding-based routing remains a competitive default, and the choice of routing method appears less impactful than other design factors we identify, such as partitioning granularity and merging interference (Q9–Q11). We will incorporate these results into the revised manuscript.
>
> **References:**
> - Ostapenko et al., "Towards Modular LLMs by Building and Reusing a Library of LoRAs," 2024. https://arxiv.org/abs/2405.11157
> - Fleshman & Van Durme, "SpectR: Dynamically Composing LM Experts with Spectral Routing," 2025. https://arxiv.org/abs/2504.03454
> - Fleshman & Van Durme, "LAG: Lazily Aggregated Gaussians for Fast Spectral LoRA Routing," 2025. https://arxiv.org/abs/2507.05346
>
> ---
>
> ## W2. Some questions target well-known LoRA characteristics
>
> We thank the reviewer for this thoughtful observation. We agree that some findings align with existing intuitions about LoRA, and we included them mainly for completeness and context. Our goal was to provide a systematic picture and to situate the less obvious findings more clearly. We will make this framing clearer in the revision.
>
> ---
>
> ## W3. Avenues for increased originality
>
> We thank the reviewer for these constructive suggestions. Both directions—synthesizing our findings into a concrete multi-LoRA system and developing information-theoretic capacity bounds or scaling laws—are exciting avenues for future work. We hope the operational boundaries mapped here provide a useful foundation for both directions.
>
> ---
>
> ## Q1. Hyperparameter tuning per experiment
>
> Thank you for this important methodological question. We detail the hyperparameter settings for each experiment in Appendix D (Table 3). For the rank sweep experiments (Q1–Q3, Q8), we set α = r so that the scaling factor adapts with rank, while learning rate and training steps are held fixed (1,500 steps, batch size 8) to isolate the effect of rank itself. For the model scale experiment (Q6), we performed a per-model hyperparameter search over learning rates (η ∈ {1e-5, 5e-5, 1e-4, 5e-4}) and training steps (S ∈ {250, 500, 1000, 2000}), with the selected configurations reported in Table 5 of the appendix.

---

> > ### Author Rebuttal · Reviewer_aR9x · 2026-03-31
> >
> > I thank the authors for addressing all of my concerns and questions. Specifically they:
> >
> > - Added comparisons with token-level methods (W1)
> > - Acknowledged the minor point about the significance of a subset of the findings. While the authors suggest they can make the framing clearer, I'd offer that this is unnecessary and agree with their sentiment that the obvious results are valuable for completeness (W2).
> > - Agree that the suggestions for improved originality are good targets for future work (W3).
> > - Answered the question on hyperparameters and pointed me to the appendix covering those details.

---

> > > ### Author Response · Authors · 2026-04-06
> > >
> > > We appreciate the reviewer's constructive engagement throughout the discussion and the confirmation that our responses have addressed all concerns. We especially value the suggestion on token-level routing methods (W1), which led to informative additional experiments that will strengthen the final manuscript.

---

### Official Review · Reviewer_T8gZ · 2026-03-08

**Soundness:** 3
**Presentation:** 3
**Significance:** 2
**Originality:** 2
**Overall Recommendation:** 4
**Confidence:** 3

**Summary:**

This paper is an experimental analysis study in which the author empirically examines 15 critical issues concerning the use of LoRA for memory in large language models and provides several insights. I acknowledge its insightful value, but I remain skeptical about whether this paper's contributions meet ICML standards, as it does not propose any substantial innovative architecture or methodology.

**Compliance With Llm Reviewing Policy:**

Affirmed.

**Final Justification:**

Most of my concerns have been solved and based on the other reviewers comments i change my score to 4.

**Key Questions For Authors:**

Frankly speaking, I find it difficult to assess the value of empirical research at high-level conferences like ICML. If this paper serves as a community guide for engineers to reference, it would be a very good piece. However, from an innovation perspective, the contributions here are somewhat weak. Many of the issues can be easily verified by engineers through straightforward methods.I would consider raising the score if other reviewers believe such empirical research is important for ICML.
1.How does the paper clearly distinguish “LoRA as memory” from prior work on multi-LoRA composition, routing, and merging?
Several existing works already study how separately trained LoRA modules can be selected, combined, or merged across domains. It would help if the authors could clarify whether the main contribution here is a genuinely new problem setting or a more systematic empirical characterization of an existing line of work.
2.To what extent do the main conclusions depend on the specific benchmarks and supervision formats used in the paper?
The results suggest that synthetic supervision format strongly affects how well knowledge is internalized into LoRA. Could the authors discuss which findings they believe are general properties of LoRA-based memory, and which may be more specific to the chosen benchmarks or data construction pipeline?
3.Is there any theoretical basis to support these experimental conclusions?

**Limitations:**

Yes.

**Strengths And Weaknesses:**

Soundness. The paper is empirically thorough and the main claims are generally well supported by series of experiments, including rank sweeps, capacity studies,  and long-context case studies. A weakness is that the work is almost entirely empirical, so several conclusions remain descriptive rather than mechanistic.

Presentation. The paper is well organized around clearly stated research questions, which makes a long empirical study easier to follow than it otherwise would be. However, the framing sometimes overstates LoRA as a “memory” abstraction before fully justifying what kind of memory is actually being measured.

Significance. The topic is important because modular knowledge updating is a real bottleneck for LLM deployment, and the paper provides practical guidance rather than just another pipeline. Still, the contribution is more diagnostic than transformative, and the practical impact depends heavily on whether the reported findings transfer to larger and more realistic continual-update settings.

Originality. The main novelty lies in the systematic, memory-centric characterization of LoRA across single-module, multi-module, and hybrid settings. Lora as a memory is more of a gimmick, as many of the 15 questions raised in this paper are already conclusions drawn from current Lora research, merely rephrased in a different way.

---

> ### Author Rebuttal · Authors · 2026-03-31
>
> We sincerely thank the reviewer for the thoughtful evaluation and openness to re-assessment.
>
> ---
>
> ## Value and originality of an empirical study at ICML
>
> We appreciate your candid assessment and the recognition that our work provides valuable practical guidance. We would like to emphasize that our contribution — a systematic empirical analysis proposing no new method — is both appropriate for ICML and substantively different from restating known conclusions.
>
> We find it reassuring that **purely empirical studies proposing no new method have strong precedent at top venues:**
>
> - [A Hitchhiker's Guide to Scaling Law Estimation](https://arxiv.org/abs/2410.11840) (ICML 2025)
> - [Learning Dynamics in Continual Pre-Training for Large Language Models](https://arxiv.org/abs/2505.07796) (ICML 2025 Oral)
> - [A Mechanistic Understanding of Alignment Algorithms: DPO and Toxicity](https://arxiv.org/abs/2401.01967) (ICML 2024 Oral)
> - [Physics of Language Models: Part 3.1, Knowledge Storage and Extraction](https://arxiv.org/abs/2309.14316) (ICML 2024)
> - [When Attention Sink Emerges in Language Models: An Empirical View](https://arxiv.org/abs/2410.10781) (ICLR 2025 Spotlight)
>
> **Our findings go beyond restating known conclusions.** We agree that some individual observations align with general intuitions — and as you note, their inclusion provides comprehensive coverage. That said, several findings went beyond what prior work could predict:
>
> - **Non-monotonic parameter efficiency** (Q3): peak efficiency at low ranks, not high.
> - **Format mixing yields superlinear gains** (Q5): complementary knowledge encoding not previously characterized.
> - **99.8% vs. 6.2% under matched parameters** (Q8): not established in prior multi-LoRA work.
> - **Practical routing negates multi-LoRA gains** (Q9): a critical warning absent from the literature.
>
> **Distinction from Prior Work.** We appreciate your question about distinction from prior work. Few recent works utilize LoRA for knowledge, but they focus on system-level performance rather than isolating LoRA's intrinsic memory properties. Consequently, critical mechanisms like capacity limits, saturation behaviors, and failure modes remain unexplored. Our contribution bridges this gap by systematically characterizing LoRA as a parametric memory, providing the fundamental understanding that prior system-building papers require but do not offer.
>
> **The other reviewers assess our contributions favorably.** Reviewer aR9x (score: 5, confidence: 5) states the paper *"addresses important questions driving active research areas"* with *"practical takeaways immediately useful to ongoing and future research,"* and highlights that the study *"comprehensively covers important aspects commonly underexplored in the literature."* Reviewer jo2J (score: 5, confidence: 4) rates both soundness and significance as "excellent." Both value systematic empirical characterization as the core contribution, and we hope this provides additional perspective on the venue-appropriateness of our work.
>
> ---
>
> ## Soundness and the "memory" framing
>
> We apologize that the scope of "memory" was not clearly delineated. Our paper studies LoRA's ability to internalize new knowledge broadly — from simple factual associations (PhoneBook) to document-level comprehension requiring multi-hop reasoning (PaperQA, NarrativeQA, QuALITY). Our descriptive approach is deliberate: before exploring how LoRA stores knowledge mechanistically, we must first establish what it can store, when it saturates, and how it fails. We will sharpen this framing in the revised Introduction.
>
> ---
>
> ## Benchmark dependence
>
> We thank the reviewer for this thoughtful question. We believe most of our main findings reflect general trends within the scope of our study, as they appear consistently across the benchmarks and model families we evaluate. At the same time, validating them on more datasets and a broader range of LLMs would strengthen this conclusion. The main qualitative trends seem more likely to generalize, while the exact format effects and quantitative gaps may depend more on the benchmark and data construction pipeline.
>
> ---
>
> ## Theoretical basis
>
> We thank the reviewer for this important question. We agree that a stronger theoretical understanding of LoRA-based memory would be valuable. As our title indicates, we position this work as an empirical analysis: rather than proposing a formal theory, our goal is to systematically characterize the empirical behavior of LoRA-based memory across rank, partitioning, merging, and hybrid settings.

---

> > ### Author Rebuttal · Reviewer_T8gZ · 2026-04-01
> >
> > Most of my concerns have been solved and based on the other reviewers comments i change my score to 4.

---

> > > ### Author Response · Authors · 2026-04-06
> > >
> > > We sincerely thank the reviewer for the constructive engagement throughout the discussion, for the positive reassessment, and for raising the score. We will sharpen the "memory" framing in the Introduction as discussed in the camera-ready version.

---

### Official Review · Reviewer_tNye · 2026-03-08

**Soundness:** 2
**Presentation:** 3
**Significance:** 2
**Originality:** 2
**Overall Recommendation:** 3
**Confidence:** 4

**Summary:**

This paper investigates the feasibility of using Low-Rank Adaptation (LoRA) as a modular, parametric knowledge memory for Large Language Models (LLMs). The authors conduct an extensive empirical study covering single-LoRA capacity scaling, the impact of synthetic data formatting (e.g., QA pairs vs. raw text), and the challenges of routing and merging in multi-LoRA systems. They evaluate their framework on synthetic datasets (PhoneBook) and mid-length reading comprehension benchmarks (NarrativeQA and QuALITY), ultimately proposing a hybrid architecture that combines merged LoRAs with non-parametric external contexts (ICL/RAG).

**Compliance With Llm Reviewing Policy:**

Affirmed.

**Final Justification:**

I thank the authors for their extensive rebuttal efforts, but my core concern remains: the proposed paradigm is ultimately a heavy-handed engineering workaround. Training multiple LoRAs and dynamically merging them to bypass gradient conflicts introduces severe I/O latency and high computational costs, making it highly impractical compared to standard RAG or ICL. While it serves as a thorough diagnostic report, it lacks the architectural elegance and real-world viability expected at this level, so my recommendation remains conservative.

**Key Questions For Authors:**

1. From a linear algebra perspective, how do you justify the claim that merging multiple small LoRAs provides a better capacity/efficiency trade-off than a single large LoRA, given that the rank of summed matrices is subadditive? Is the multi-LoRA advantage merely an artifact of avoiding gradient conflict during training?

2. Table 2 shows that on the QuALITY dataset, pure ICL (70.20) outperforms Multi-LoRA Top3 + ICL (69.22) for the Llama-3.1-8B model. Does this not indicate that the merged low-rank matrices act as noise that disrupts the model's native in-context processing?

3. Given that standard RAG vastly outperforms pure Multi-LoRA on QuALITY (63.79 vs. 42.42) without the training costs and merging latency, what is the practical justification for adopting the LoRA-as-memory paradigm in real-world engineering?

4. Have you evaluated this framework on modern, extremely long-context benchmarks (e.g., LongBench v2) to verify if the routing and merging mechanics can survive massive document lengths?

**Limitations:**

yes

**Strengths And Weaknesses:**

**Strengths**:

1. Extensive Empirical Effort: The paper presents a heavy and comprehensive experimental report. The ablations on data formatting (QA vs. Summary vs. Raw) and latency breakdowns are thorough and provide a detailed look at the engineering overhead of LoRA-based memory.

2. Transparent Reporting: The authors honestly report the latency bottlenecks (I/O overhead of dynamic loading) and the performance drops associated with embedding-based routing.

**Weaknesses**:

1. Flawed Mathematical Premise on Rank Capacity: The paper argues that a single large LoRA is less efficient and more prone to saturation than merging multiple small LoRAs. This contradicts the fundamental properties of matrix rank. The sum of multiple low-rank matrices is strictly bounded (rank(A+B+C)≤rank(A)+rank(B)+rank(C)). Thus, merging three rank-4 LoRAs yields a matrix with a maximum rank of 12, which is theoretically subsumed by the representational space of a single rank-16 LoRA. The observed "saturation" in large LoRAs is likely a symptom of gradient interference and catastrophic forgetting during end-to-end training, not a theoretical capacity ceiling. The multi-LoRA approach merely bypasses this optimization issue via chunked, isolated training and routing, masking the underlying rank limitations.

2. Redundancy with RAG (An Over-engineered Retrieval System): The proposed pipeline (chunking the document → generating QA pairs → training multiple LoRAs → embedding-based retrieval → weight merging) is essentially a highly convoluted and computationally expensive version of RAG. It introduces massive training overhead and severe I/O latency during inference, yet fundamentally relies on the same retrieval mechanisms as standard RAG.

3. Inferior Standalone Performance and "Harmful" Hybrids: The empirical results undercut the value of the proposed architecture. On the QuALITY dataset, the pure Multi-LoRA Top3 approach scores only 42.42(LLama 3.1 8B), while standard zero-shot RAG achieves 63.79. To compensate, the authors evaluate hybrid setups. However, pure ICL achieves 70.20 on QuALITY, whereas Multi-LoRA Top3 + ICL drops to 69.22. This suggests that injecting merged, interfering LoRA weights actually contaminates the model's clean representation space, degrading its native in-context reasoning capabilities.

4. Format Overfitting vs. Genuine Comprehension: By heavily relying on GPT-generated QA pairs for training, it remains highly ambiguous whether the LoRA module is truly internalizing complex, long-context semantic relationships, or if it is merely overfitting to the specific QA formats and stylistic patterns of the synthetic training data.

5. Lack of Rigorous Long-Context Benchmarks: NarrativeQA and QuALITY (typically <20k tokens) do not adequately represent the complexities of modern long-context reasoning. The paper lacks evaluations on rigorous, complex benchmarks like LongBench v2 or ∞-Bench, where multi-hop reasoning across 100k+ tokens would likely cause the multi-LoRA merging strategy to collapse completely.

---

> ### Author Rebuttal · Authors · 2026-03-31
>
> We thank the reviewer for the detailed engagement. We group related concerns for coherence.
>
> ---
>
> ## W1&Q1. Multi-LoRA capacity claim contradicts rank subadditivity
>
> We thank the reviewer for this important point. We agree that rank subadditivity holds. However, this is not the mechanism behind our main multi-LoRA result.
>
> The concern seems to interpret Q8 as a **merging** setup. In fact, Q8 is a **routing** setup: each query activates exactly one LoRA. Since weights are not summed, rank subadditivity does not apply. The advantage comes not from exceeding a single LoRA's rank budget, but from partitioning knowledge across modules.
>
> Our merging experiments (Q10--Q11) are also consistent with the reviewer’s intuition: when multiple LoRAs are actually merged, performance degrades relative to routing.
>
> ---
>
> ## W2&W3&Q2&Q3. LoRA is redundant with RAG and harmful in the complex hybrid setups
>
> **(a) Our paper does not propose LoRA as a RAG replacement.** Our Introduction states the objective as a "systematic audit of LoRA as a parametric memory"; our Conclusion calls LoRA "best viewed not as a replacement for context-based methods, but as a complementary component." The QuALITY gap (63.79 vs. 42.42) is a closed-book, standalone setting — precisely the limitation we identify and diagnose. Our contribution is mapping when LoRA helps and when it fails.
>
> **(b) LoRA and RAG are fundamentally different mechanisms.** RAG retrieves *text* into the context window at every query; multi-LoRA routing selects *parameters*, consuming zero context tokens. Q15 shows LoRA achieves a lower processing time under repeated querying.
>
> **\(c) It is very slightly harmful only in 1/8 settings but beneficial 7/8 settings.**
>
> We find it difficult to conclude that hybrid LoRA is harmful when it improves over ICL in 7 of 8 settings, with only one slightly worse case.
>
>
> | Model | NQA (Hybrid vs. ICL) | QuALITY (Hybrid vs. ICL) |
> |---|---|---|
> | Llama-1B | 26.41 vs. 24.52 (**+1.89**) | 30.00 vs. 26.01 (**+3.99**) |
> | Llama-8B | 38.78 vs. 33.81 (**+4.97**) | 69.22 vs. 70.20 (−0.98) |
> | Qwen-1.7B | 29.17 vs. 27.34 (**+1.83**) | 56.52 vs. 46.80 (**+9.72**) |
> | Qwen-8B | 37.23 vs. 35.22 (**+2.01**) | 76.26 vs. 72.89 (**+3.37**) |
>
> Hybrid LoRA improves over standalone ICL in 7 of 8 settings. The overall pattern is more consistent with complementarity than with systematic degradation. Additional 100K+ token experiments (see W5&Q4) further confirm this complementarity.
>
> ---
>
> ## W4. LoRA may overfit to the synthetic QA format rather than genuinely internalizing knowledge
>
> This is a valid concern for any synthetic-data-driven study. We believe several pieces of evidence argue against pure format-overfitting.
>
> **PaperQA's 3-level hierarchy tests beyond format matching.** PaperQA spans three cognitive levels (Appendix C), where Level 3 questions require deducing a paper's logical flow — unsolvable by reproducing QA templates.
>
> **All synthetic formats improve over raw text, not just QA.** Figure 4 shows Summary and Rewrite also outperform raw text despite no structural resemblance to QA evaluation. The key factor is knowledge density, not format alignment.
>
> **Format mixing outperforms QA alone (Q5).** If LoRA were overfitting to QA patterns, adding non-QA formats should dilute performance. Instead, the combined configuration achieves the best score (6.822 vs. 5.893 for QA alone on Llama), suggesting complementary knowledge encoding.
>
> **Long-context QA transfer.** LoRA modules trained with synthetic data are evaluated on NQA and QuALITY, whose question formats differ entirely from GPT-generated training data. Consistent improvements in these out-of-distribution settings (Table 2) further counter the overfitting hypothesis.
>
> ---
>
> ## W5&Q4. Lack of evaluation on 100k+ token benchmarks
>
> We thank the reviewer for this suggestion. We ran additional experiments on both benchmarks with documents exceeding 100K tokens.
>
> **InfiniteBench** (ROUGE-L, 20 documents ≥100K tokens):
>
> | Method | Qwen-8B | Qwen-1.7B |
> |---|---|---|
> | ICL | 30.89 | 14.30 |
> | RAG | 32.80 | 30.46 |
> | Single LoRA | 24.37 | 17.91 |
> | Multi-LoRA Top3 | 23.49 | 20.41 |
> | Multi-LoRA Top3 + ICL | **34.92** | 29.60 |
> | Multi-LoRA Top3 + RAG | 32.10 | **31.39** |
>
> **LongBench v2** (Accuracy, 30 documents, 101K–364K tokens):
>
> | Method | Qwen-8B | Qwen-1.7B |
> |---|---|---|
> | ICL | 20.00 | 20.00 |
> | RAG | 20.00 | 16.67 |
> | Single LoRA | 30.00 | 23.33 |
> | Multi-LoRA Top3 | 30.00 | **36.67** |
> | Multi-LoRA Top3 + ICL | **33.33** | 26.67 |
> | Multi-LoRA Top3 + RAG | 16.67 | 30.00 |
>
> Contrary to the concern, LoRA-based methods do not collapse at 100K+ tokens but remain competitive, reinforcing our findings: hybrid outperforms either alone (∞Bench), and in extreme-length regimes where context-based methods struggle, parametric memory offers a distinct advantage (LongBench v2).

---

> > ### Author Rebuttal · Reviewer_tNye · 2026-04-01
> >
> > I appreciate the authors' extensive efforts during the rebuttal, particularly the impressive undertaking of evaluating on LongBench v2 and InfiniteBench. The new results demonstrating performance gains in the 100k+ token regime effectively address my concerns regarding the framework's survivability in extreme long-context scenarios. Furthermore, I value the authors' candid agreement on the mathematical limitations of rank subadditivity and their clarification that the benefits arise primarily from partitioning rather than expanded representational capacity.
> >
> > In light of these comprehensive empirical additions, I am raising my score. **However**, I align closely with Reviewer T8gZ: while this is an exhaustive and practically useful diagnostic report, the fundamental engineering paradigm— **dynamically loading and merging disk-bound LoRA weights to serve as an extremely high-latency, computationally heavy**  RAG alternative—still feels more like a heavy-handed workaround to circumvent gradient conflicts, rather than a fundamentally elegant or scalable architectural solution for long-context memory.

---

> > > ### Author Response · Authors · 2026-04-06
> > >
> > > We are grateful to the reviewer for the thoughtful re-evaluation and for raising the score. We are glad that the additional 100K+ experiments and the clarification on rank subadditivity were helpful.
> > >
> > > We appreciate the candid remark on the engineering elegance of the current multi-LoRA paradigm. We share the view that dynamic loading and merging is not the end state — rather, we hope our diagnostic study serves as a useful empirical foundation for future architectural work toward more principled solutions.

---

### Official Review · Reviewer_jo2J · 2026-03-12

**Soundness:** 4
**Presentation:** 3
**Significance:** 4
**Originality:** 3
**Overall Recommendation:** 5
**Confidence:** 4

**Summary:**

The paper explores empirically how LoRA behaves as knowledge storage, as opposed to task adaptation, which is what LoRA is traditionally used for. LoRA's behavior as knowledge storage is still an open question despite the fact that many prior works have used LoRA for this specific application before. The experiment aims to study LoRA's intrinsic behavior and failure modes as knowledge storage. The paper finds that LoRA can catastrophically fail in certain settings (cross-chunk synthesis, sub-optimal routing), resulting in practical guidance for practitioners. Also, the paper shows that LoRA can be used as a complementary module on top of established RAG and ICL approaches. Finally, the paper introduces two new benchmarks, PhoneBook and PaperQA, designed for probing memory storage.

**Compliance With Llm Reviewing Policy:**

Affirmed.

**Final Justification:**

I do believe that the paper provides substantial understanding of how LoRAs (and parameters in general) store knowledge in LLMs. It is a thorough empirical investigation that shows various insights and can be used to improve future generations of PEFT techniques and continual learning.

**Key Questions For Authors:**

1. Do the merging methods in Q10 operate with the low-rank matrices (A and B) or the full-weight matrices?
2. Can the authors give some plausible explanation why CAT merging is much worse than linear? Have the authors tried CAT with magnitude scaling each LoRA matrix with $\sqrt{N}$, where $N$ is the number of LoRAs to be merged (this would be equivalent to linear average of the corresponding full-weight matrices)
3. Are the LoRAs in Q15 merged to the base model before inference? What is the ready time in Fig. 8? How much time is spent on routing for dynamic multi-LoRA setting?

**Limitations:**

yes

**Strengths And Weaknesses:**

Strengths
- The research topic is relevant and timely given that prior works have been using LoRA as knowledge storage with no concrete understanding of its behavior. It also has implication for long-term knowledge and continual learning which current foundation models are still lacking.
- Experiment setup is thorough aiming the answer various practical questions related to LoRA's ability to serve as parametric knowledge storage.
- The empirical results are surprisingly thorough and dense, providing a practical guideline for using LoRA as knowledge storage in practical settings.


Weaknesses
- Since the empirical analysis is very dense, the experimentation descriptions in some places are not elaborate enough to understand in the first pass. For instance, the setup for Q11 is confusing since the oracle is used for routing but more than one LoRA are retrieved. In Q14, it is not clear what "contextual continuity" means in this context. I believe cutting a few results from or condensing the conclusion in the main text would significantly help in terms of clarity.

---

> ### Author Rebuttal · Authors · 2026-03-31
>
> We sincerely thank the reviewer for the thorough and constructive evaluation. We address each point below.
>
> ---
>
> ## W1. Clarity of dense experimental descriptions (Q11 setup, Q14 terminology)
>
> Thank you for the careful reading. We clarify both points below.
>
> **Q11 setup.** We apologize for the confusion. Q11 does not involve routing — it isolates **merge-induced interference**. We select N documents, merge their N ground-truth LoRA modules, and evaluate on queries drawn from all N documents. Varying N from 1 to 5, we measure how co-merging additional (all correct) modules affects performance. The monotonic degradation confirms that interference, not missing knowledge, is the primary bottleneck. We will revise the text to make this distinction explicit.
>
> **Q14 "contextual continuity."** We agree this term was insufficiently defined. It refers to cross-chunk semantic dependencies severed when a long document is partitioned into separate LoRA modules — individual modules hold only partial views, hindering multi-hop reasoning across chunk boundaries. We will add an explicit definition in the revision.
>
> ---
>
> ## Q1. Do merging methods operate on low-rank (A, B) or full-weight matrices?
>
> **All merging methods in Q10 operate on the low-rank matrices (A and B), not on the reconstructed full-weight delta.** Specifically:
>
> -  **Linear averaging** averages each of A and B separately across modules: $A_{\text{merged}} = \sum_i \lambda_i A_i$, $B_{\text{merged}} = \sum_i \lambda_i B_i$.
> -  **TIES and DARE** apply their trim/sign-election/rescaling steps independently to A and B.
> -  **CAT** concatenates A along one dimension and B along another, as described in Appendix O.
>
> This follows the standard approach of each method and the HuggingFace PEFT implementation.
>
> ---
>
> ## Q2. Why is CAT much worse than linear? Would scaling by 1/√N help?
>
> We are grateful for this excellent suggestion, which directly led to a new and informative experiment.
>
> **Why vanilla CAT fails.** The primary cause is scale mismatch. Without normalization, CAT computes the *sum* of individual deltas ($\Delta W = \sum_i B_i A_i$), whereas Linear computes the *mean* ($\Delta W = \frac{1}{N} \sum_i B_i A_i$). For $N=3$, this results in a $3\times$ larger perturbation to the base model, severely distorting the output distribution.
>
> **Scaled CAT ($1/\sqrt{N}$).** As the reviewer identified, scaling each $A_i$ and $B_i$ by $1/\sqrt{N}$ yields the linear average of full-weight deltas while retaining the expanded rank $Nr$. We ran this experiment:
>
> | Method              | Llama-3.1-8B | Qwen3-8B |
> | ------------------- | :----------: | :------: |
> | CAT                 |     0.19     |   2.30   |
> | CAT ($1/\sqrt{N}$)  |     4.77     |   5.41   |
> | Linear              |     4.92     |   5.76   |
> | TIES                |   **5.41**   | **5.87** |
>
> The $1/\sqrt{N}$ scaling dramatically recovers CAT performance, confirming scale mismatch as the dominant failure mode. The remaining gap to Linear and TIES reflects parameter interference — conflicting updates that scaling alone cannot resolve, unlike TIES, which explicitly mitigates them via sign election. We will include this analysis in the revision. We thank the reviewer again for the suggestion that motivated this experiment.
>
> ---
>
> ## Q3. Clarifying the latency accounting in adapter application, ready time, and routing overhead
>
> We thank the reviewer for these questions. We believe the main issue is that the latency accounting in Q15 and Fig. 8 was not clearly defined. We clarify below which costs are included and where the overhead in multi-LoRA actually comes from.
>
> **Are the LoRAs in Q15 merged to the base model before inference?** No. They are not merged into the base model as a one-time step ahead of inference. In Q15, the adapters are handled at query time: for each query, the relevant LoRAs are selected, merged into a temporary adapter, and then used for generation.
>
> **What does "ready time" mean in Fig. 8?** It refers to the one-time setup cost before processing any query. This includes base model loading; for retrieval-based methods, it additionally includes RAG initialization (embedding model loading, chunk embedding, FAISS index construction). We agree that this term was not clear and will clarify this in the revision.
>
> **How much time is spent on routing in the dynamic multi-LoRA setting?** Routing takes about 8 ms per query, small relative to the other latency components. The dominant overhead in the dynamic multi-LoRA is adapter loading from disk, which is why dynamic loading appears as the main bottleneck in Fig. 8. In the preloaded setting, the remaining overhead mainly comes from per-query adapter merging. We will add the routing number explicitly in the revision, since Fig. 8 currently presents the main latency categories but does not break them out.

---

> > ### Author Rebuttal · Reviewer_jo2J · 2026-04-01
> >
> > I thank the authors for their response. I have no further questions. I'm happy to keep a positive score. I encourage the authors to add clarifications in the camera-ready version, which has one additional page.

---

> > > ### Author Response · Authors · 2026-04-06
> > >
> > > We sincerely thank the reviewer for confirming that our responses have addressed all concerns. We are grateful for the constructive suggestions throughout the review process — in particular, the √N-scaled CAT experiment (Q2), which yielded a genuinely informative result that strengthens the paper.
> > >
> > > As encouraged, we will incorporate all clarifications (Q11 setup, Q14 terminology, Fig. 8 labeling, and the scaled CAT analysis) into the camera-ready version, making full use of the additional page.

---

### Decision · Program_Chairs · 2026-04-30

**Decision:**

Accept (regular)

**Comment:**

This paper provides a thorough and timely empirical investigation into using LoRA as a parametric knowledge memory, addressing an important gap in understanding its capacity, composability, and failure modes. Across multiple reviewers, there is strong agreement on the technical soundness, extensive experimentation, and practical value of the findings, with several reviewers highlighting that the study offers actionable guidance for both research and real-world deployment. Concerns regarding theoretical framing, engineering overhead, and limited architectural novelty are valid but have been substantially clarified in the rebuttal, particularly the distinction between routing and merging, the role of LoRA as a complementary rather than replacement mechanism to RAG/ICL, and the additional long-context evaluations. While the work is primarily diagnostic rather than proposing a new method, the depth, consistency, and newly surfaced empirical insights make it a meaningful contribution that is likely to inform future advances in modular and continual learning for LLMs. Overall, I recommend acceptance.